# An exploratory modelling study of perennial firn aquifers in the Antarctic Peninsula for the period 1979-2016.

J. Melchior van Wessem[1], Christian R. Steger[2], Nander Wever[3], and Michiel R. van den Broeke[1]

[1]Institute for Marine and Atmospheric Research Utrecht, Utrecht University, Utrecht, the Netherlands
[2]Institute for Atmospheric and Climate Science, ETH Zürich, Zürich, Switzerland
[3]Department of Atmospheric and Oceanic Sciences, University of Colorado Boulder, Boulder, CO, USA

*Correspondence to:* J. M. van Wessem (j.m.vanwessem@uu.nl)

**Abstract.**

In this study, we focus on the model detection in the Antarctic Peninsula (AP) of so-called perennial firn aquifers (PFAs), that are widespread in Greenland and Svalbard and are formed when surface meltwater percolates into the firnpack in summer, is then buried by snowfall, and does not refreeze during the following winter. We use two snow models, the IMAU Firn

Densification Model (IMAU-FDM) and SNOWPACK and force these (partly) with mass and energy fluxes from the Regional Atmospheric Climate MOdel (RACMO2.3p2) to construct a 1979-2016 climatology of AP firn density, temperature and liquid water content. An evaluation using 75 snow temperature observations at 10 m depth and density profiles from 11 firn cores, shows that output of both snow models is sufficiently realistic to warrant further analysis of firn characteristics.

The models give comparable results: in 941 model grid points in either model, covering $\sim$28,000 km$^2$, PFAs existed for

at least one year in the simulated period, most notably in the western AP. At these locations, surface meltwater production typically exceeds 200 mm w.e. y$^{-1}$, with accumulation for most locations >1000 mm w.e. y$^{-1}$.

Most persistent and extensive are PFAs modelled on and around Wilkins ice shelf. Here, both meltwater production and accumulation rates are sufficiently high to sustain a PFA on 49% of the ice shelf area, in (up to) 100% (depending on the model) of the years in the 1979–2016 period. Although this PFA presence is confirmed by recent observations, its extent in the models

appears underestimated. Other notable PFA locations are Wordie ice shelf, an ice shelf that has almost completely disappeared in recent decades, and the relatively warm northwestern side of mountain ranges in Palmer Land, where accumulations rates can be extremely high and PFAs are formed frequently. PFAs are not necessarily more frequent in areas with the largest melt and accumulation rates, but they do grow larger and retain more meltwater, which could increase the likelihood of ice shelf hydrofracturing.

We find that not only the magnitude of melt and accumulation is important, but also the timing of precipitation events relative to melt events. Large accumulation events that occur in the months following an above average summer melt event, favour PFA formation in that year. Most PFAs are predicted near the grounding lines of the (former) Prince Gustav, Wilkins and Wordie ice shelves. This highlights the need to further investigate how PFAs may impact ice shelf disintegration events through the process of hydrofracturing, in a similar way as supraglacial lakes do.

# 1 Introduction

The Antarctic Peninsula (AP) is one of the most rapidly changing regions on Earth. In the second half of the 20th century, the lower atmosphere over the AP has warmed up to 3K, which is among the highest warming rates globally (Bromwich et al., 2012). At the same time, heat content in the surrounding ocean (Schmidtko et al., 2014) and precipitation rates have increased (Thomas et al., 2008), and in the western AP most perennial sea ice has disappeared (Stammerjohn et al., 2011). Largely as a result of these changes, 90% of its glaciers are retreating and ice shelves on both sides of the AP have disintegrated or have shown area loss and thinning (Cook et al., 2014, 2016). Rising atmospheric temperatures increase surface meltwater production, which has been linked to ice shelf disintegration, as it depletes firn air content and can hydrofracture the ice shelf when the meltwater fills pre-existing crevasses (Scambos et al., 2000; Van den Broeke, 2005). It is therefore pivotal to correctly model the fate of meltwater when it enters the firn.

For long it has been assumed that meltwater in Antarctica either refreezes immediately in the firn, runs off directly into the ocean, or forms supraglacial lakes (melt ponds) on the ice shelf surface. However, liquid meltwater inside the firn can also survive winter until the next melt season without refreezing. These so-called perennial firn aquifers (PFA) are extensive on the Greenland ice sheet where they were first discovered by Forster et al. (2013). They have been further studied during subsequent field campaigns, both using in situ (e.g. Koenig et al., 2014; Miller et al., 2017), seismic (e.g. Montgomery et al., 2017), and airborne/satellite measurements (e.g. Miège et al., 2016; Brangers et al., 2020; Miller et al., 2020). PFAs reduce and delay runoff and impact processes related to ice shelf hydrofracturing by storing meltwater in the firn before it percolates into crevasses (Poinar et al., 2017; Bell et al., 2018; Alley et al., 2018). PFAs modulate surface meltwater transport to the ice sheet base where it has a lubricating effect (Kingslake et al., 2015; Poinar et al., 2017; Bell et al., 2018), or affect both grounded ice and ice shelves through latent heat release if eventually refrozen (Fountain and Walder, 1998; Hubbard et al., 2016).

Recently, a PFA has been found on the Wilkins ice shelf (Montgomery et al., 2020), and others studies discuss meltwater buried by firn or ice but these are not technically called aquifers (Lenaerts et al., 2017a; Dunmire et al., 2020). In Greenland, PFAs are found in regions which combine high accumulation with high surface melting rates (Kuipers Munneke et al., 2014a). When accumulation, representing precipitation minus sublimation, is sufficiently large it can bury and isolate the summer meltwater from the winter cold wave, preventing refreezing. Kuipers Munneke et al. (2014a) suggested optimal PFA conditions to be accumulation rates between 1000 and 3000 $mm$ w.e. $y^{-1}$. These conditions prevail in the (western) AP, where snowfall rates are generally high due to orographic precipitation. Additionally, the AP is the northernmost and therefore warmest region in Antarctica, with significant surface melting in summer (Van Wessem et al., 2016) and regular pronounced foehn-induced melt events even in winter (Kuipers Munneke et al., 2018). Its climate is therefore relatively similar to that of southeast Greenland, where extensive PFAs have been observed as well as modelled (Forster et al., 2013; Steger et al., 2017a; Brangers et al., 2020).

In this study, AP firn characteristics including potential PFA formation are assessed using two snow models: IMAU-FDM (Ligtenberg et al., 2011) and SNOWPACK (Bartelt and Lehning, 2002). The models are forced by realistic atmospheric and surface conditions from the regional climate model RACMO2.3p2 for the period 1979–2016, which has been extensively evaluated with observational datasets, as have previous versions (Van Wessem et al., 2015, 2016, 2018), and includes a snow

model that is physically identical to IMAU-FDM (Ettema et al., 2010; Ligtenberg et al., 2011). This combination of models accurately simulates PFA locations in Greenland (Forster et al., 2013; Steger et al., 2017a). Section 2 introduces the model and observational datasets used for evaluation. Section 3 evaluates the model output using available observations of firn density and temperature, while section 4 discusses the differences between SNOWPACK and IMAU-FDM. Section 5 treats the specific PFA characteristics of the AP. Section 6 provides a discussion of results and future prospects of meltwater modelling in the AP, followed by conclusions in Section 7.

## 2 Data and Methods

Two snow models are used, the Institute for Marine and Atmospheric research Utrecht firn densification model (IMAU-FDM) (Ligtenberg et al., 2011) and SNOWPACK (Bartelt and Lehning, 2002; Lehning et al., 2002; Keenan et al., 2020). A detailed description of differences in model architecture is provided in Steger et al. (2017a). Output from the RACMO2.3p2 regional atmospheric climate model is used to force these models offline. These models are described below.

### 2.1 Regional Atmospheric Climate Model RACMO2.3

The Regional Atmospheric Climate Model RACMO2 has been extensively used and evaluated for climatological studies of the ice sheet of Greenland and Antarctica, the latest model version being RACMO2.3p2 (Noël et al., 2018; Van Wessem et al., 2018). In this study, the model has been applied at a relatively high horizontal resolution of ∼5.5 km over the AP and is forced by ERA-Interim re-analysis data (Dee et al., 2011). The model atmosphere is initialised on Jan. 1st, 1979 using ERA-Interim. RACMO2.3p2 includes a 100-layer firn model that calculates percolation, refreezing and runoff of liquid water (Ettema et al., 2010). The output of this internal firn model is not used in this study as the model is physically identical to IMAU-FDM described below, but the latter runs at a higher vertical resolution (100 layers in RACMO2 versus 3000 layers in IMAU-FDM). RACMO2.3p2 3-hourly data are used to force IMAU-FDM and SNOWPACK at all land ice grid points (see Fig. 1), but in a different way for both models as will be described in Section 2.3. Further details about RACMO2.3p2 and the evaluations in the AP are provided in Van Wessem et al. (2015, 2016, 2018).

### 2.2 IMAU-FDM

This semi-empirical model was described in detail in Ligtenberg et al. (2011), and this section only repeats the main characteristics. The model consists of 3000 layers of varying thickness, of which the properties (temperature, mass, density and liquid water content) are followed in a Lagrangian way through the firn pack. The vertical water motion is simulated using the bucket scheme, in which meltwater moves through all layers within a single three hourly model time step; water is retained as ice or liquid water based on the firn porosity and cold content. The irreducible water content is set to a relatively low constant value of 2% of the pore volume, compared to the temperature dependent ∼4% in SNOWPACK and in other studies (Coléou and Lesaffre, 1998; Lafaysse et al., 2017), allowing meltwater to efficiently percolate down to lower layers, mimicking processes such as piping and meltwater retention. The model only allows meltwater to be retained through the irreducible water content

and does not allow water ponding on superimposed ice; if liquid water reaches the firn-ice boundary, it is assumed to run off directly. The FDM is forced using three-hourly fields from 1979 to 2016 of accumulation (snowfall minus sublimation), snow erosion, 10 meter wind speed, surface temperature and snowmelt from RACMO2.3p2 (Van Wessem et al., 2018). The lower firn boundary condition for the heat equation is set as a Neumann zero-flux boundary condition, which is a good assumption in the deep firn where gradients are expected to be negligibly small (Ligtenberg et al., 2011).The initial firn pack is generated by forcing the FDM with as many climatological periods (1979-2016) as needed to refresh the entire firn layer. The spin up time depends on the depth of the firn layer, which in turn depends on the average accumulation and snowmelt at the respective grid point, and therefore varies spatially. The final initialisation product is in near-balance with the average climate, and provides the best estimate of the initial state of the AP firn pack.

## 2.3 SNOWPACK

SNOWPACK is a microphysical snow model originally designed to model seasonal snow cover in alpine areas (Bartelt and Lehning, 2002; Lehning et al., 2002). It has recently been applied successfully to the Antarctic (Keenan et al., 2020) and Greenland ice sheets (Steger et al., 2017b). In the latter study, a simplified version of SNOWPACK using a bucket scheme for meltwater percolation was used to reduce computational costs and to allow comparison with IMAU-FDM. Water ponding is not allowed in the version used here, and meltwater retention is simulated using an irreducible water content depending on temperature, and averages 4% by pore volume. For further details we refer to Steger et al. (2017a, b).

In this study, besides using a new version of SNOWPACK, the main change from the Greenland simulation is the forcing mode. We found that when the model calculates its own surface energy budget (SEB), instead of using direct forcing with surface temperature, this yielded better results for the AP. To that end, three-hourly fields from RACMO2.3p2 for 1979–2016 of precipitation, downwelling and upwelling shortwave- and longwave radiation fluxes, 10 m wind speed, 2 m air temperature and specific humidity are used (Van Wessem et al., 2018). As both the up- and downward shortwave fluxes are prescribed, this automatically sets the surface snow albedo. The turbulent (sensible and latent heat) fluxes are based on the bulk method and are calculated using near-surface temperature and humidity gradients. The SEB components are then used to calculate sublimation/deposition, evaporation/condensation and melt, and thus result in slightly different values than the surface fluxes from RACMO2 used to force IMAU-FDM (quantified in Sec. 4). As in IMAU-FDM, the lower firn boundary condition for the heat equation is set as a Neumann zero-flux boundary condition. The model is spun up in a fashion similar to the IMAU-FDM, by forcing the model with as many climatological periods (1979-2016) as needed to refresh the entire firn layer. In general, densification in SNOWPACK is slightly weaker than in IMAU-FDM (Steger et al., 2017a) and as a result SNOWPACK spinup for some cold and dry locations does not refresh the full firn layer. The effects on the results of this study are very small. To further decrease computational costs (and storage), SNOWPACK uses a variable number of layers in combination with aggressive layer merging scheme, which merges layers when they are thin and when their physical properties are similar. There are some datagaps in the SNOWPACK simulations due to model crashes partly as a result of these thin snow layers in dry and cold, high elevation regions, but these are situated outside the areas discussed in this study.

## 2.4 Observational data

### 2.4.1 10 m firn temperature

This study uses 10 m firn temperatures from the Surface Mass Balance and Snow Depth on Sea Ice Working Group (SUMup) (Montgomery et al., 2018) for model evaluation (locations in Fig. 1). All observations applicable to this study have unfortu-
nately been obtained prior to the starting year of the simulations (1979, as atmospheric reanalyses are unreliable before 1979 (Dee et al., 2011)) and cannot directly be matched with the model results; hence the climatological average from 1979–2016 is used for modelled firn temperature. To investigate the potential impact of the non-overlapping periods of observations and model, we resorted to data from two stations from the SCAR-READER dataset (Turner et al., 2004) that have temperature observations from before 1979, one representing the western AP (Faraday/Vernadsky) and one the eastern AP (Marambio).
Annual modelled temperature for 1979–2016 agrees within 0.35 K. The station temperatures are 1.2 K and 0.5 K lower for the 1950-1978 period, compared to the 1979-2016 period, a change representative of the warming in the second half of the 20th century. We thus conclude that the generally underestimated model temperatures (see next section Fig. 2) cannot be (partly) ascribed to the non overlapping period.

Measurements that are not located on a RACMO2 land-ice grid point or that are located on ice shelves that have disintegrated before 2008 are not included, resulting in 75 temperature observations that can be used for evaluation. All observations are corrected for any discrepancies in model elevation using a temperature lapse rate of $-7.2 \text{ K km}^{-1}$ (Morris and Vaughan, 2003).

### 2.4.2 Density profiles from Larsen C

Modelled density profiles are evaluated using observations from 11 firn cores from the Larsen C ice shelf (Tab. 1, Fig. 1). Six shallow to medium-deep density observations as reported in Munneke et al. (2017) are used. J1, J2 and J3 reach up to 30 m depth and were collected in 2009 using a neutron-scattering probe with a vertical resolution of 10 cm. LAR1, LAR2 and LAR3 are 6 m deep firn drillings, also from the 2009 field season. The other five observations are deep (100 m depth) firn density profiles collected with optical televiewing (OPTV) of hot-water drilled boreholes as described in Ashmore et al. (2017), with 1 cm vertical resolution.

## 3 Results: model evaluation

Figure 2 compares observed with modelled SNOWPACK (blue) and IMAU-FDM (red) 10 m firn temperature $T_{10m}$. SNOW-PACK and IMAU-FDM simulate snow temperatures with comparable skill: $r^2$= 0.91; 0.85, RMSD = 1.55;1.89 K and slope = 0.90; 0.80, respectively. SNOWPACK performs best at simulating the higher temperatures, found in locations with summer melt and significant firn warming due to refreezing. Both models on average underestimate 10 m snow temperature, which is partly related to known errors in the atmospheric forcing; i.e. underestimated downwelling longwave and shortwave radiation. These biases also largely explain the underestimation in surface melt production in both models (Ligtenberg et al., 2018; Van Wessem et al., 2018). There are only limited observations with $T_{10m} > 265$ K; here melt rates are high ($> 500 \text{ mm w.e. y}^{-1}$)

and as a result the firn layer is either shallow, or absent. At these locations, the modelled $T_{10m}$ represents ice temperature. There are also a few outliers for both models; these represent locations on steep slopes, likely with complex atmospheric and surface conditions that are poorly resolved by the model.

Figure 3 compares modelled and observed density profiles on Larsen C ice shelf, with statistics in Table 2. The shallow cores in the upper panel show relatively good agreement. SNOWPACK typically models less dense profiles than IMAU-FDM, e.g. cores J1_08, LAR1 and LAR3, where SNOWPACK/IMAU-FDM mostly underestimate/overestimate observed density slightly. As the (output) vertical resolution of SNOWPACK in our setting is coarse, due to the aggressive layer merging necessary for optimisation of computation time, IMAU-FDM shows larger variability with depth. For the deeper cores, which reach considerably higher densities than the shallow cores, the IMAU-FDM is closer to the observations, especially for the CI-cores, simulating several layers of ice that qualitatively agree with the observations. For the two WI cores, both models deviate strongly from the observations, especially in the upper 40 m, where the observations represent a vertical profile that almost completely consists of ice.

## 4    Results: model intercomparison

A detailed description of differences in model structure is provided in Steger et al. (2017a). As described in Sec. 2, the surface forcing of both models is different. For instance, where IMAU-FDM uses sublimation and meltwater production rates prescribed by RACMO2, SNOWPACK calculates its own surface energy budget and therefore sublimation and melt rates. Figure 4 shows the major firn climatologies with a focus on the intermodel differences.

Figures 4a-c show the modelled FAC of IMAU-FDM (a) and SNOWPACK (b) and the difference (c). The calculated FAC is shown for the first 30 meters only, disregarding the deeper layers, enabling a more useful comparison for this application as meltwater percolation mostly affects the upper firn. The overall pattern is similar in both models. In warmer regions with sustained surface melting (see Figs. 4g,h), i.e. the Larsen C, George VI and Wilkins ice shelves, FAC is low, with values approaching 1 m. In these regions, SNOWPACK simulates larger FAC than the IMAU-FDM. In regions with large FAC (i.e. large accumulation rates and/or little melt), such as the western mountain range of Palmer Land and several high elevation locations to the south, SNOWPACK generates lower FAC values. This reflects the faster dry snow densification in IMAU-FDM.

Figures 4d-f compare simulated 10 m firn temperature, illustrating the generally warmer firn conditions simulated by SNOW-PACK. This is a rather uniform pattern over the AP, with only some locations where IMAU-FDM $T_{10m}$ is higher. Figs. 4g-i show that melt rates are largely similar. IMAU-FDM generally predicts somewhat larger values ($\sim25$ mm w.e. y$^{-1}$) than SNOWPACK, except over George VI and Wilkins ice shelves, and near the former Prince Gustav ice shelf. These differences in melt are explained by differences in the turbulent heat fluxes (not shown), but overall the differences are small, i.e. less than 10%. Finally, Figs. 4j-l show the average vertically integrated liquid water content (LWC), which represents the meltwater that is not refrozen or does not runoff in the same model timestep. The patterns of LWC are similar to those of meltwater production in both models: LWC is mainly concentrated on the ice shelves and towards the north. Values are generally lower than the

surface meltwater availability, as most meltwater refreezes, even in summer. This is clearest on Larsen C ice shelf, where only a small fraction of meltwater remains liquid. Note that these figures do not represent perennial firn aquifers (PFAs); for these, liquid water must still be present after winter. These locations are discussed in Section 5.

The LWC differences between the two models do not always coincide with those in surface melt rates. For instance, over some parts of Wilkins ice shelf, where absolute values of LWC peak, SNOWPACK LWC can be smaller than in IMAU-FDM, while the melt is actually higher. In most regions, as melt rates but also firn porosity and firn depth are larger, LWC values in SNOWPACK are larger, but in some regions with high melt rates, the firn is much thinner as most of it is melted away, and LWC values are lower.

In summary, both models behave similarly, but small differences remain. Importantly, several locations exist where both models predict that the meltwater does not refreeze or runs off to the ocean, but is retained as liquid water in the firn layer even outside the summer season, suggesting the presence of PFAs. This will be discussed in the next section.

## 5    Results: Perennial firn aquifers (PFAs)

Figure 5 shows the annual average vertically integrated LWC for those grid points with PFA presence, i.e. where liquid water is persistent throughout at least one hydrological year (October to September of the next year in the period 1979-2016), for IMAU-FDM and SNOWPACK. In total, 796 IMAU-FDM and 864 SNOWPACK model grid cells conform to this definition, with 941 grid cells having a PFA in either of the two models, which corresponds to an area of roughly 28,000 $km^2$, about 50% of the $\sim$55,000 $km^2$  PFA area observed in Greenland (Brangers et al., 2020). PFA grid cells with LWC reaching high values ($> 1000$ $kg\,m^{-2}$) occur mainly on the western slopes in northwest Palmer Land, where accumulation as well as melt rates are relatively high. The two models largely agree in most areas, but IMAU-FDM shows larger LWC values. The Wilkins ice shelf stands out as the main PFA region, with PFAs predicted over 49% of its area, a peak in LWC in the northernmost part of the ice shelf, and an additional peak in Schubert inlet. Towards the ice shelf interior, where surface melting is weaker, LWC decreases.

Smaller PFAs are predicted by both models along the western AP from Wordie ice shelf towards the northern tip. In many of these locations, both models predict LWC values >1000 $kg\,m^{-2}$. In the eastern AP the models simulate PFAs in a few isolated locations on the grounded ice near the remnants of the Larsen A and Prince Gustav ice shelves, with low LWC values. The Larsen B embayment shows a significant difference between the models; IMAU-FDM does not predict any PFAs here, whereas SNOWPACK does. Here, LWC values are low (<30 $mm$ w.e. $y^{-1}$), PFAs usually only last one year and water is not found deeper than 10 m (not shown). The difference between the two models is mainly explained by the difference in irreducible water content; in IMAU-FDM the meltwater is vertically distributed more quickly, after which it refreezes more easily.

Figure 6 shows the persistence of the PFAs as a fraction of the total 37 year period (1979–2016) covered by the model simulations. For the Larsen B embayment, the small PFAs are only present during <5 % ($< 2$ years) of the time. For the other PFAs the pattern reflects that of LWC: in some locations the PFA is present during the entire period, but for most locations the PFAs occur intermittently: PFAs are not formed during cold periods with less melt because all liquid water refreezes in the firn.

Figure 7 shows time series of the yearly PFA area, and the yearly average melt (in both models) and accumulation for these grid points. Note that IMAU-FDM and SNOWPACK snowmelt rates are very similar, albeit SNOWPACK values are slightly higher (~5%) in some years.

The largest PFA area is modelled in the 1990s, with a peak of ~19,000 km$^2$ in (in IMAU-FDM) in 1998. During the last decade, snowmelt has decreased, which is related to atmospheric cooling over the AP (Van Wessem et al., 2015; Turner et al., 2016), and PFA area diminishes as liquid water gradually freezes. PFA area in IMAU-FDM is ~20% larger than in SNOWPACK (not shown). This difference is largely explained by Wilkins ice shelf. The temporal variability in both models is similar.

In Fig. 7 ,a clear relation between PFA extent and annual averages of accumulation or snowmelt is not obvious, but periods of PFA area growth do occur when average melt exceeds ~400 mm w.e. y$^{-1}$, or about 25% of the annual average accumulation, such as in 1986-1989 or 2005-2006. It is also interesting that SNOWPACK simulates PFAs over a larger number of different grid points (864 vs 796) from year to year, while in every individual year IMAU-FDM has a larger total surface area (and total LWC) (see Fig. 7); apparently, SNOWPACK models some PFAs at variable locations that only last a single year, while IMAU-FDM does not, see e.g. the small PFAs in the Larsen A and Larsen B embayments. It is clear that PFA formation is complex and depends on specific melt and accumulation conditions which are different for each location. Therefore a more detailed analysis will be provided below.

The subtle interplay between surface meltwater production and accumulation in PFA formation is illustrated in Figures 8 and 9. This analysis follows Kuipers Munneke et al. (2014b) who found that in an idealised experiment PFAs form in relatively warm locations with sufficient surface meltwater production, but also require high snow accumulation rates. Figures 8a,b show the distribution of RACMO2 grid points that fall in certain yearly melt and accumulation ranges, for both IMAU-FDM and SNOWPACK. The vast majority of grid points falls in the lower left corner, with modest melt and accumulation rates. Numbers quickly drop for larger melt and accumulation rates, and the slightly higher melt rates in SNOWPACK (Fig. 7) are reflected by additional grid points at high melt rates. Figures 8c,d show the distribution for PFA grid points and years; i.e. with liquid water in the firn during at least one full hydrological year. Numbers are now much lower; a peak of >100 points is found in both models at a melt rate of ~400 mm w.e. y$^{-1}$ and an accumulation rate of ~750 mm w.e. y$^{-1}$. Figures 8e,f show the PFA occurence fraction, i.e. the PFA count (c,d) divided by the total amount of model grid points (a,b). PFA formation is clearly favoured by high melt and accumulation towards the top right corner, but at the same time these conditions are relatively rare.

Figure 9 highlights conditions favourable for PFA formation, by showing the same distribution as in Figs. 8c,d, but now only for years in which a PFA is first formed or has increased in size. This means that years in which a PFA shrinks are not considered, even though the PFA may still be present. Figures 9a,b show that PFA formation/growth is restricted to higher melt and accumulation rates compared to PFA presence (Fig. 8); PFAs only (significantly) form/grow at melt rates >150 mm w.e. y$^{-1}$ and for accumulation rates of >500 mm w.e. y$^{-1}$. Extending the criteria as before to PFA fraction of the total number of grid points (Fig. 9c,d) no distinct optimum is seen, apart from a small band of higher values around a melt rate of 550 mm w.e. y$^{-1}$. Towards the top right, conditions are found with the highest fractions of PFA formation, but the number of

points is small. Both models simulate roughly the same PFA behaviour, but for SNOWPACK maxima are found for a larger number of conditions, and the larger melt rates in SNOWPACK result in more grid points for these bins.

Finally, Figures 9e,f show the fraction of summer meltwater that is retained in the firn layer at the end of winter, only when this fraction is positive and above 5%. This highlights the efficiency of the firn to prevent refreezing of subsurface meltwater. Even though the pattern is similar to Figs. 9c,d, there is a clear preference for very high accumulation: at higher accumulation rates, the fraction of meltwater that is retained increases, favouring PFA formation in subsequent years. This also implies that were melt and accumulation rates to increase in the future, PFA are likely to expand. This will be the topic of a forthcoming study.

## 5.1 Results: Case studies at three aquifer locations

Three specific PFA locations are selected for case studies. The locations and their climatic conditions are shown in Figs. 1, 5 and 8, respectively.

### 5.1.1 Wilkins ice shelf

In both models, Wilkins ice shelf (WIS) has the most prominent PFA signature in the AP. The presence of liquid water in the firn on WIS is confirmed by satellite observations (Scambos et al., 2009), and recently an extensive PFA has been on the ice shelf (Montgomery et al., 2020). Figures 8,9 identify WIS as a site with favourable conditions in terms of accumulation and surface melt water production. This is also a region where significant supraglacial water (melt ponds; Scambos et al., 2009) and saturated firn (Alley et al., 2018) have been observed. Figure 10 shows time series of volumetric LWC profiles from SNOWPACK and IMAU-FDM for a location on WIS with annual LWC $>500 \, \mathrm{kg \, m^{-2}}$ (Fig. 5). This particular location is not necessarily representative of the entire WIS PFA system, which is variable in water content and retention depth, but constitutes a PFA with large interannual variability and is therefore most interesting for a case study. At this location PFAs are formed in both models in several years. The most significant ones, where more than 5% of the meltwater is retained, are highlighted with grey bars in Fig. 10. 1984/1985 is the first year in which meltwater penetrates to $\sim$10 m depth in both models, and is retained throughout the entire following year, but then refreezes the next year. PFA formation at WIS is most significant in the relatively warm 1990-1999 period, during which the PFA is growing and extending deeper in the firn pack, lasting until 2005 in SNOWPACK, and 2009 in IMAU-FDM. After that, as temperatures and melt decrease (Van Wessem et al., 2015), the liquid water refreezes at this site.

The presence of an extensive PFA on Wilkins ice shelf (WIS) as reported by Montgomery et al. (2020) is confirmed by our results. Although they do not provide a quantitative estimate of total surface area of the WIS aquifer, their results from the MCoRDS radar system onboard NASA's Operation IceBridge fight on 16 November 2014 suggest a greater westward aquifer extent than in either IMAU-FDM or SNOWPACK (Fig. 5). In addition, their study provides field observations of e.g. liquid water content and firn density. A comparison of these observations with our results remains difficult, as these in-situ observations were performed in 2018, i.e. after the model period used here. Moreover, neither firn model treats meltwater

ponding on top of ice lenses or lateral water flow, further hampering such a direct comparison. It is well possible that the underestimated PFA extent is a result of these model limitations.

Figure 10c shows local monthly accumulation and melt from RACMO2. In years with above average melt rates (horizontal lines that represent the mean plus one standard deviation thresholds) enough meltwater is available to initiate or grow a PFA.

### 5.1.2  Wordie ice shelf

When compared with its surface area in the 1960s, only 10% of Wordie ice shelf remains in 2008, with the largest area decrease in the 1990s (Cook et al., 2014). This makes an interesting case to explore the potential impact of PFA formation on ice shelf stability. Figure 5 shows PFAs near the grounding line, with considerably larger LWC in IMAU-FDM than in SNOWPACK. Figure 11 shows the volumetric LWC for one of these locations. IMAU-FDM simulates a PFA over the whole period, growing significantly from 2002 onwards and extending even below 50 m depth, while SNOWPACK only shows a marginal aquifer in the late 1990s and early 2000s.

In Fig. 11c, the monthly melt and accumulation rates do not show a clear formation threshold in melt and/or accumulation, but the combination of high melt with high accumulation afterwards is visible in some years.

### 5.1.3  Palmer Land

Apart from Wilkins ice shelf, the most robust presence of PFAs in both models is in the northwest AP mountain range in Palmer Land. Here accumulation rates in RACMO2 reach up to several meters water equivalent per year. Being located far north and without much sea ice cover year-round, this area also experiences considerable surface melting. Figure 5 identifies many PFAs, frequently with LWC values above $>1000$ kg m$^{-2}$, mostly over the islands and the (north)west facing slopes. Time series for one of these locations are shown in Figure 12, with an averag accumulation rate of 5.1 m w.e. y$^{-1}$ and a relatively low melt rate of 240 mm w.e. y$^{-1}$(see Fig. 8). Nearly every year up to 2008 a PFA develops, but not many are retained longer than about 1-2 years. This is caused by the extreme accumulation rates that deeply bury the summer meltwater, as evident from the steep slopes of LWC fraction with respect to time. Because most snow falls at subzero temperatures, these deep snow layers have sufficient cold content to refreeze the meltwater from above and below. The strong firn compaction caused by the high accumulation rates also causes a reduction of the amount of liquid water that can be retained, which results in some meltwater runoff, which is otherwise rare in Antarctica. This behaviour is seen for most PFA locations in this region that have melt rates below approximately 300 mm w.e. y$^{-1}$. In regions with significantly higher melt rates but similar accumulations rates, such as the islands, PFAs are found that almost completely fill the firnpack, resulting in large LWC values in both models (Fig. 5). Note that in Palmer Land, LWC in both firn models is comparable.

Fig. 12c highlights the extremely large accumulation rates, which are an order of magnitude larger than melt. As accumulation is so large, it is most of the time sufficient to isolate the summer meltwater. Only in years where both accumulation and melt are relatively low, no PFAs are formed.

## 6 Discussion

### 6.1 Intermodel differences

This model study shows the significant potential for the presence of perennial firn aquifers (PFAs) in the AP. They are predicted to occur mostly in the western AP where both melt and accumulation rates are large. The results are largely coherent between the IMAU-FDM and SNOWPACK firn models, but in some regions, most notably near the former Wordie ice shelf (Figs. 5 and 11), large differences were found. On average IMAU-FDM simulates deeper PFAs and higher LWC values at most PFA locations (Fig. 5). Locations were also found where SNOWPACK predicted (small) PFAs, while IMAU-FDM did not (Sec. 5), such as the Larsen A and B ice shelf embayments. Such intermodel differences can be mainly ascribed to differences in the irreducible water content, but also by differences in surface melting, snow densification parameterizations and the firn pack initialization; the accumulation forcing for both models is equal.

IMAU-FDM has a fixed irreducible water content of 2%, substantially lower than (the snow temperature dependent) irreducible water content in SNOWPACK, which averages ∼4%. As a result, in IMAU-FDM water percolates to greater depths quicker, where it either refreezes or runs off (Steger et al., 2017a). Exceptions are locations where melt rates are sufficiently high to saturate the whole firn column, e.g. in the northern Wilkins ice shelf and the islands in the northwestern AP. On the one hand the lower irreducible water content of IMAU-FDM allows meltwater to spread out deeper into the firn, where it can more efficiently refreeze or runoff, such as in regions such as the Larsen A and B ice shelf embayments, while in SNOWPACK some meltwater still remains in the upper layers. On the other hand, in regions with moderate melt rates such as Wordie ice shelf, the larger irreducible water content in SNOWPACK likely causes more meltwater to be retained in the upper layers, where it can be more efficiently refrozen by the winter cold wave, resulting in the much smaller PFAs present. Therefore, differences in the representation of irreducible water are important but also subtle, depending on the interplay of local firn and atmospheric conditions.

Additionally, the spinup of both models is different. The spin-up time of IMAU-FDM is made dependent on the yearly average snowfall and surface meltwater production, so as to rebuild the entire firn layer. For SNOWPACK the same approach is used, but, with firn densification being weaker in SNOWPACK, the spin up does not everywhere replace the entire firn column. Longer spinups were not performed due to computational costs. This results in some of the SNOWPACK initial firn profiles to be colder than in IMAU-FDM, having more potential for refreezing. However, this only affects a few locations, and mostly in dry and cold regions where PFAs do not form.

### 6.2 Firn air content, lateral meltwater transport and superimposed ice formation

Section 3 showed large discrepancies between models and observations for some locations on Larsen C ice shelf: modelled densities are too low (and firn air content too high) for cores WI-0 and WI-70, which may apply to other regions with large melt rates as well, such as the embayments of Larsen A and Larsen B ice shelves. Combined with both models being, on average, too cold (Fig. 2), this potentially reflects the lack of lateral meltwater flow in the snow models, which would result in locally more refreezing, more latent heat, higher temperatures and higher densities. Hubbard et al. (2016) reported on deep/thick ice layers

in the firn and related these to gentle depressions in surface topography, causing meltwater to accumulate from surrounding locations, which also explains the extensive melt ponds (Kingslake et al., 2017). Future research should focus on incorporating these effects in firn models, for instance using Alpine 3D (Lehning et al., 2006) which allows lateral movement of meltwater, using more sophisticated physical models (Buzzard et al., 2018), or incorporating processes that affect meltwater percolation, such as (horizontal) piping (Reijmer et al., 2012), preferential flow (Wever et al., 2016) or subsurface radiation penetration and melting (van Dalum et al., 2018). Many of the PFAs in this study are found near the grounding line, often on grounded ice, so that horizontal meltwater transport can also transport PFA meltwater towards the ice shelves with additional implications for their stability (Lenaerts et al., 2017b).

On the other hand, Holland et al. (2011) and Kuipers Munneke et al. (2014b) suggested that modelled firn air content on large parts of Larsen C ice shelf is too low, caused by either meltrates that are overestimated, or accumulation rates that are too low, or both. These discrepancies can also be related to how deep the meltwater penetrates into the firn and how important the sensitivity of the melt-albedo feedback is (Jakobs et al., 2018), because this determines to a large extent how much meltwater is produced and the potential to fill pore space in the firn. Obviously, to tackle these problems, more observational constraints are needed.

Finally, both firn models currently lack a realistic physical representation of meltwater retention: they only retain meltwater in the firn due to irreducible water. In reality, meltwater is also retained due to the ponding on top of superimposed ice (slush), similar to the formation of melt ponds. As a result, a direct comparison with observed water volume is currently not viable, and the current approach should be regarded as an exploratory study of firn processes. Nonetheless, because irreducible water that does not refreeze is a requirement for PFA formation, aquifer location, formation and horizontal extent can still be predicted using these simplified models (Ligtenberg et al., 2011; Forster et al., 2013; Steger et al., 2017a).

### 6.3 PFA seasonality

Sections 5.1.1-5.1.3 showed the evolution of PFAs at three locations. It is clear that for PFA formation accumulation and melt rates should be high enough, but the monthly time series shown in Figures 10-12c suggest that the timing of accumulation could also play a role. Figure 13 shows the seasonality of monthly melt and accumulation for the extreme PFA formation years (solid lines) where more than 25% of meltwater is retained at the end of winter, representing the highest values melt/accumulation conditions in Figs. 9e,f. When compared to the average for all years, it appears that aport from above-normal summer melt, accumulation is significantly larger ($>1\sigma$) in April during efficient PFA formation. It should be noted that this result is an average over multiple PFA locations, which ignores that PFA formation is a subtle process in which many factors, including the timing and seasonality of accumulation, play a role. Investigating this further is beyond the scope of this exploratory study.

### 6.4 Ice shelves

Hydrofracturing has been implied in the breakup of Larsen A, Larsen B and Prince Gustav ice shelves (Scambos et al., 2000; Van den Broeke, 2005; Bell et al., 2018) as well as in the recent breakup of large parts of Wilkins ice shelf (Scambos et al., 2009). Ice-shelf breakup has typically been linked to meltwater ponding that provides a source of water to drain into surface

crevasses. Subsurface water from PFAs might also be a source for hydrofracturing. A careful look at Figure 5 reveals that, especially for SNOWPACK, PFA locations on grounded ice frequently coincide with the grounding lines of ice shelves that have partly disintegrated, or with locations that no longer have ice shelves. Especially the timing (late 1990s) of PFAs on Wordie ice shelf coincides with some of its larger disintegration events (Cook et al., 2014). Moreover, the remarkable longevity of the

5 Wordie ice shelf PFA (at least as suggested by IMAU-FDM (Figs. 6 and 11)) would increase the probability that lateral flow brought the meltwater to a crevassed section of the ice shelf, increasing the likelihood of hydrofracturing (Poinar et al., 2017). Other disintegrated ice shelves in the eastern AP also show the potential for PFA presence on their grounding lines, e.g. former Prince Gustav and Larsen A ice shelves. This does not imply causation and could simply be a result of warmer conditions. In future work, the potential role of PFAs on ice shelf stability will be studied in more detail, to gain better understanding of the

10 fate of ice shelves in a warming climate in which both melt and snowfall, and thus PFA formation, are expected to increase.

## 7   Conclusions

Two snow models, the IMAU Firn Densification Model (IMAU-FDM) and SNOWPACK, are used to assess firn characteristics of Antarctic Peninsula (AP) glaciers, with a specific focus on predicting the presence of perennial firn aquifers (PFAs). PFAs represent surface meltwater that percolates into the firnpack in summer and remains liquid after being buried by snowfall during

the following fall/winter. PFAs have the potential to modulate meltwater runoff (Kingslake et al., 2015), or to, like melt ponds, impact ice shelf stability by modifying processes of ice shelf hydrofracturing and subsequent break-up (Poinar et al., 2017; Bell et al., 2018; Banwell et al., 2019). The two snow models are forced with output from the Regional Atmospheric Climate MOdel (RACMO2.3p2) at 5.5 km horizontal resolution, to construct a 1979–2016 climatology of AP firn density, temperature and liquid water content. A comparison with 75 snow temperature observations at 10 m depth and with density from 11 firn

cores suggests that both snow models perform adequately. Both models tend to underestimate 10 m firn temperature (RMSD $\sim$1.5K) but the spatial variability is represented well ($r^2 \sim 0.90$). SNOWPACK performs better in warmer locations with larger surface melt rates, which is important for the formation of PFAs. Both models tend to overestimate the amount of firn pore space, both in warm and cold locations. Previous work has shown that both models are capable of simulating PFAs in Greenland (Forster et al., 2013; Steger et al., 2017a).

The two models predict PFAs covering $\sim$28,000 $\mathrm{km^2}$, during at least one year during 1979-2016. Generally the estimates of PFA locations are consistent among the models. PFAs are found in regions with both significant surface melt and high accumulation rates. These conditions are met most frequently along the western coastline of the AP, but some smaller PFAs are also modelled on the eastern AP, notably near the grounding lines of former Prince Gustav, Larsen A and Larsen B ice shelves. The most extensive PFA system is modelled on Wilkins ice shelf, covering 49% of its total area in both models. Obser-

vations confirm the presence of an extensive aquifer on the WIS, but suggest that the modelled extent may be underestimated (Montgomery et al., 2020). The PFAs with highest liquid water content are found along the western AP coastline and on the adjacent islands, most significantly so in Palmer Land, where accumulation rates often exceed 3000 $\mathrm{mm}$ w.e. $\mathrm{y^{-1}}$, i.e. highly efficient to isolate summer meltwater in the firn from the winter cold. Interannual variability of PFA presence is relatively large.

Most PFAs exist longer than one year, several PFAs even persist during the whole 38 years of the model study. Most PFAs are found in the period from 1987–2000, the warmest episode. In the last two decades, PFA counts have significantly decreased, following atmospheric cooling that has been reported previously (Van Wessem et al., 2015; Turner et al., 2016).

Both high melt and accumulation rates generally favour PFA formation, but too much melt may result in a summer snowpack that melts away, and too much accumulation buries the meltwater so deep and adds so much cold content that the PFA never grows significantly. Optimal conditions for PFA formation, in which more than 50% of summer meltwater is retained over the subsequent winter, are found at locations with both high melt and accumulation rates. If atmospheric temperatures rise further in the future, increasing both surface melt and accumulation, we thus expect more meltwater will be retained in the firn and PFAs will become more prevalent and persistent.

Not only the magnitude and ratio of yearly melt and accumulation are important, but also the seasonal variability. A relatively warm summer with high melt followed by a large accumulation event in the first month of fall (April) appears to favour PFA formation. On Wilkins ice shelf, PFAs formed in many years, and meltwater bodies from different years can merge deeper in the firn causing the PFA to grow. In northwest Palmer Land, a region with extremely large accumulation rates, deep aquifers are found but they seldom merge with firn aquifers of other years. For this to happen melt rates should be considerably larger.

This study can guide observational studies to locate Antarctic PFAs. Its outcomes also suggest a link between PFA formation and ice shelf viability, as it appears PFAs on the grounded ice coincide with grounding lines of (former) ice shelves that have disappeared or are disintegrating, such as Prince Gustav, Wilkins and Wordie ice shelves. A further analysis should reveal to what extent PFAs impact ice shelf stability, and whether PFAs increase the likelihood of ice shelf disintegration through hydrofracturing (Poinar et al., 2017).

## Acknowledgements

We are grateful for the financial support of NWO/ALW, Netherlands Polar Programme. We acknowledge support by PROTECT. This work was partly funded by the NWO (Netherlands Organisation for Scientific Research) VENI grant VI.Veni.192.083.

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

**Table 1.** Density from 11 firn cores used for comparison with modelled firn density. See Munneke et al. (2017) for details of the shallow cores (upper six) and Ashmore et al. (2017) for the deeper cores. Firn core locations are shown in Fig. 1.

| Site | Latitude | Longitude | Depth (m) |
|---|---|---|---|
| J1_08 | 67.1 | 61.3 | 30 |
| J2_08 | 68.3 | 68.15 | 30 |
| J4_08 | 68.4 | 64.7 | 30 |
| LAR1 | 68.14 | 63.95 | 6 |
| LAR2 | 67.57 | 63.25 | 5 |
| LAR3 | 67.03 | 62.64 | 5 |
| CI-120 | -67.0 | -61.5 | 90 |
| CI-0 | -66.4 | -63.38 | 97.5 |
| CI-22 | -66.6 | -63.2 | 90 |
| WI-0 | -67.4 | -64.9 | 90 |
| WI-70 | -67.5 | -63.3 | 90 |

**Table 2.** Mean, bias (model – observation), root-mean-square deviation and correlation coefficient ($r^2$ with significance level $p < 0.0001$) of density profiles for the 11 firn cores described in Table 1. Denoted are also observed and modelled firn air content (FAC; in meters), and the annual average melt (M; mm w.e. $y^{-1}$) and accumulation (A;mm w.e. $y^{-1}$) forcing. Both modelled and observed density profiles are vertically interpolated on a regular vertical grid and statistics are based on this grid.

| | obs/forcing | | | IMAU-FDM | | | | SNOWPACK | | | |
|---|---|---|---|---|---|---|---|---|---|---|---|
| | FAC | M | A | FAC | bias | RMSD | $r^2$ | FAC | bias | RMSD | $r^2$ |
| J1_08 | 7.0 | 232 | 323 | 2.7 | 133 | 92 | 0.29 | 7.6 | -17 | 84 | 0.28 |
| J2_08 | 10.5 | 140 | 480 | 9.4 | 31 | 81 | 0.39 | 11.7 | -38 | 58 | 0.48 |
| J4_08 | 7.9 | 138 | 554 | 9.4 | -51 | 77 | 0.36 | 11.8 | -124 | 58 | 0.48 |
| LAR1 | 3.1 | 149 | 459 | 2.4 | 35 | 90 | 0.11 | 3.2 | -10 | 49 | 0.17 |
| LAR2 | 2.4 | 239 | 385 | 1.8 | 75 | 95 | 0.41 | 2.2 | 37 | 81 | 0.27 |
| LAR3 | 2.1 | 265 | 351 | 1.5 | 118 | 147 | 0.18 | 2.1 | 13 | 100 | 0.26 |
| CI-120 | 6.1 | 289 | 324 | 3.1 | 31 | 41 | 0.63 | 13.4 | -61 | 76 | 0.47 |
| CI-0 | 5.0 | 309 | 363 | 2.6 | 23 | 52 | 0.02 | 11.0 | -58 | 111 | 0.01 |
| CI-22 | 2.2 | 292 | 311 | 0.6 | 17 | 25 | 0.01 | 7.6 | -37 | 76 | 0.35 |
| WI-0 | 4.3 | 184 | 645 | 14.8 | -107 | 94 | 0.38 | 26.2 | -214 | 80 | 0.38 |
| WI-70 | 5.3 | 218 | 412 | 8.5 | -34 | 54 | 0.66 | 26.1 | -204 | 56 | 0.67 |

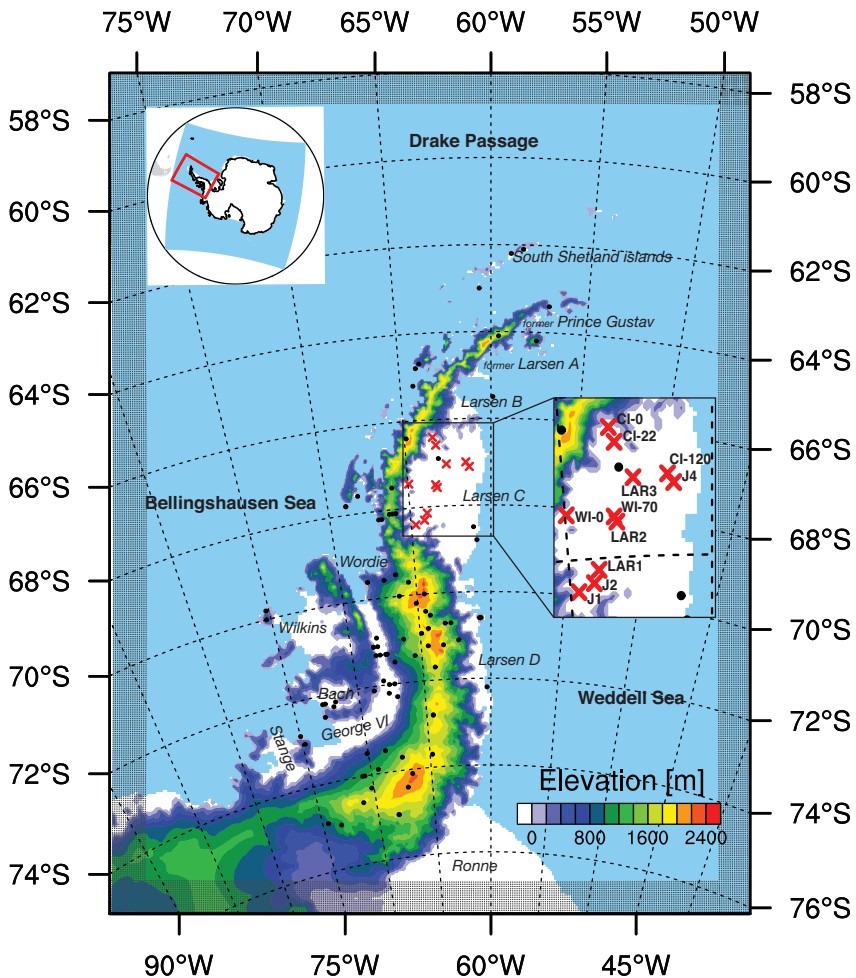

**Figure 1.** RACMO2 model domain (red box in inset map of Antarctica) and surface topography (m) of the Antarctic Peninsula. Locations of 10 m snow temperature observations are marked (black dots), as well as the density profiles (red crosses). Model topography is based on digital elevation models described in Van Wessem et al. (2018). White areas represent the floating ice shelves, coloured contours represent the grounded ice sheet.

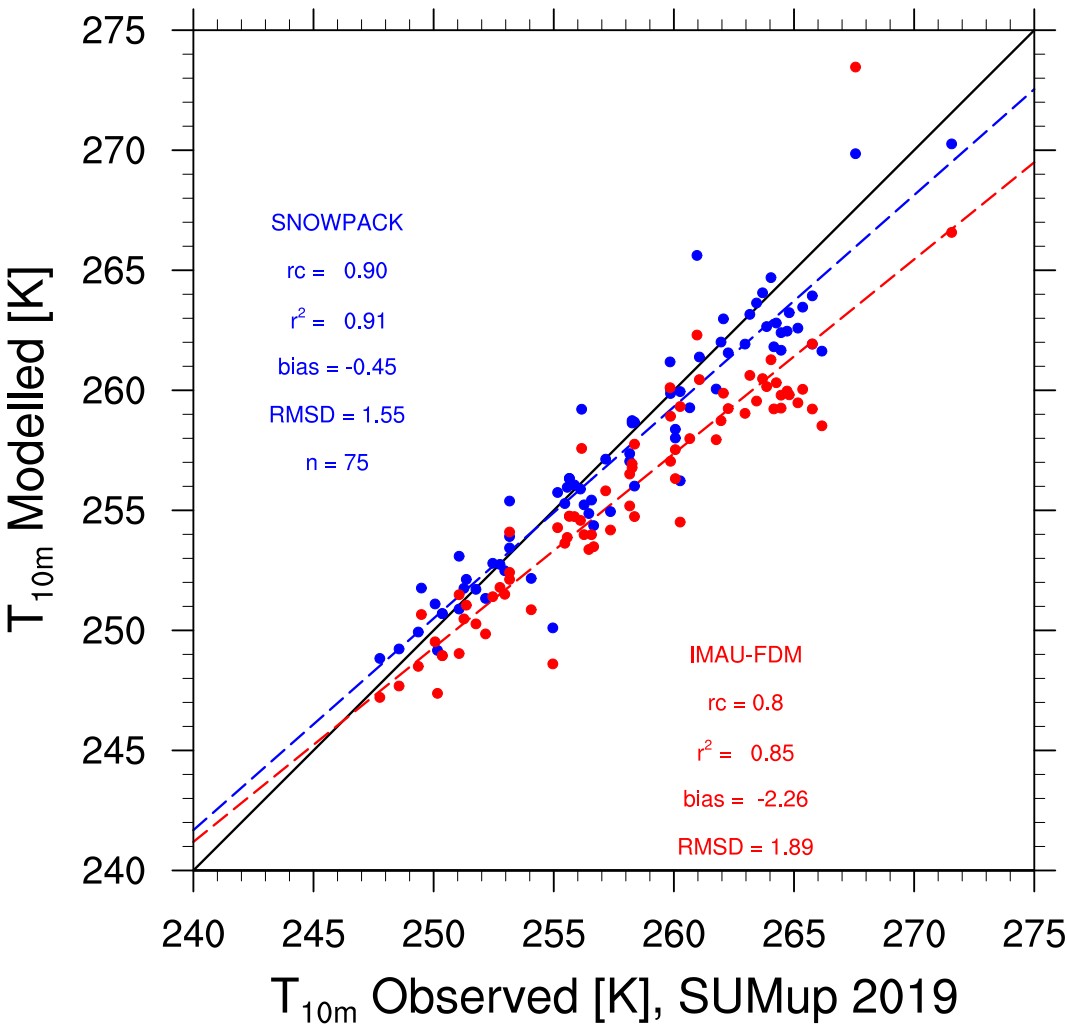

**Figure 2.** Modelled IMAU-FDM (red), SNOWPACK (blue) and RACMO (black) 10 m snow temperature as a function of observed 10 m snow temperature. Modelled temperature is corrected for discrepancies in elevation using a lapse rate of 7.2 K km$^{-1}$. Statistics (slope (rc), $r^2$ (with significance level p < 0.0001), bias and RMSD) for SNOWPACK, IMAU-FDM and RACMO-FDM are denoted based on all 75 locations. Observations are from Montgomery et al. (2018) and selected as described in Section 2.4.1.

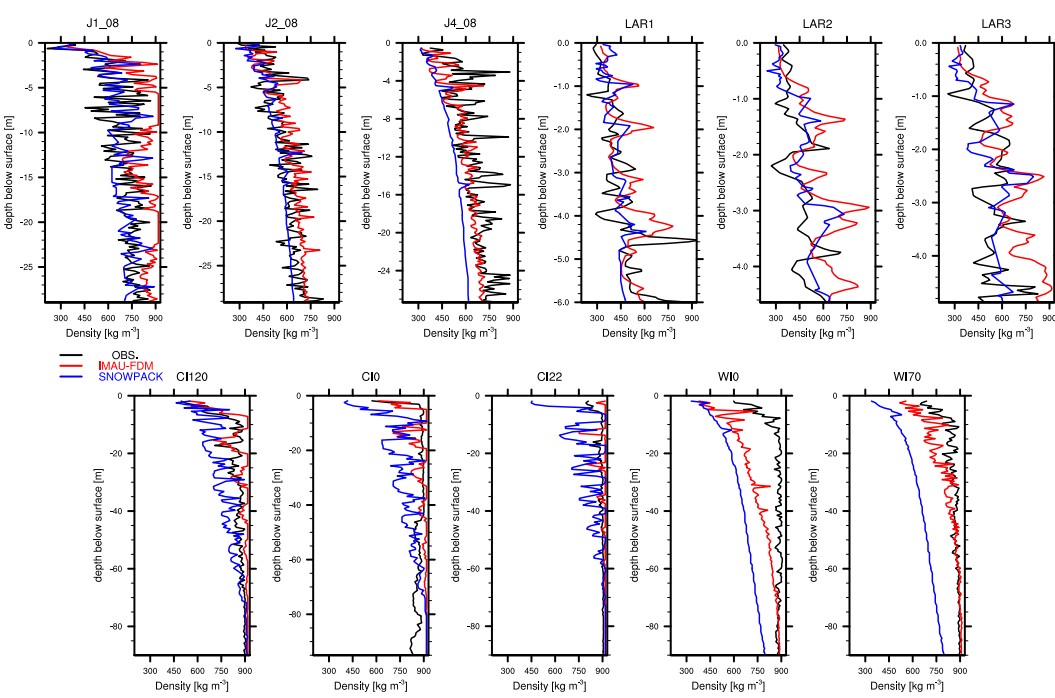

**Figure 3.** Modelled IMAU-FDM (red), SNOWPACK (blue) and observed (black) density profiles for 11 locations on the Larsen C ice shelf (Sect. 2.4.2).

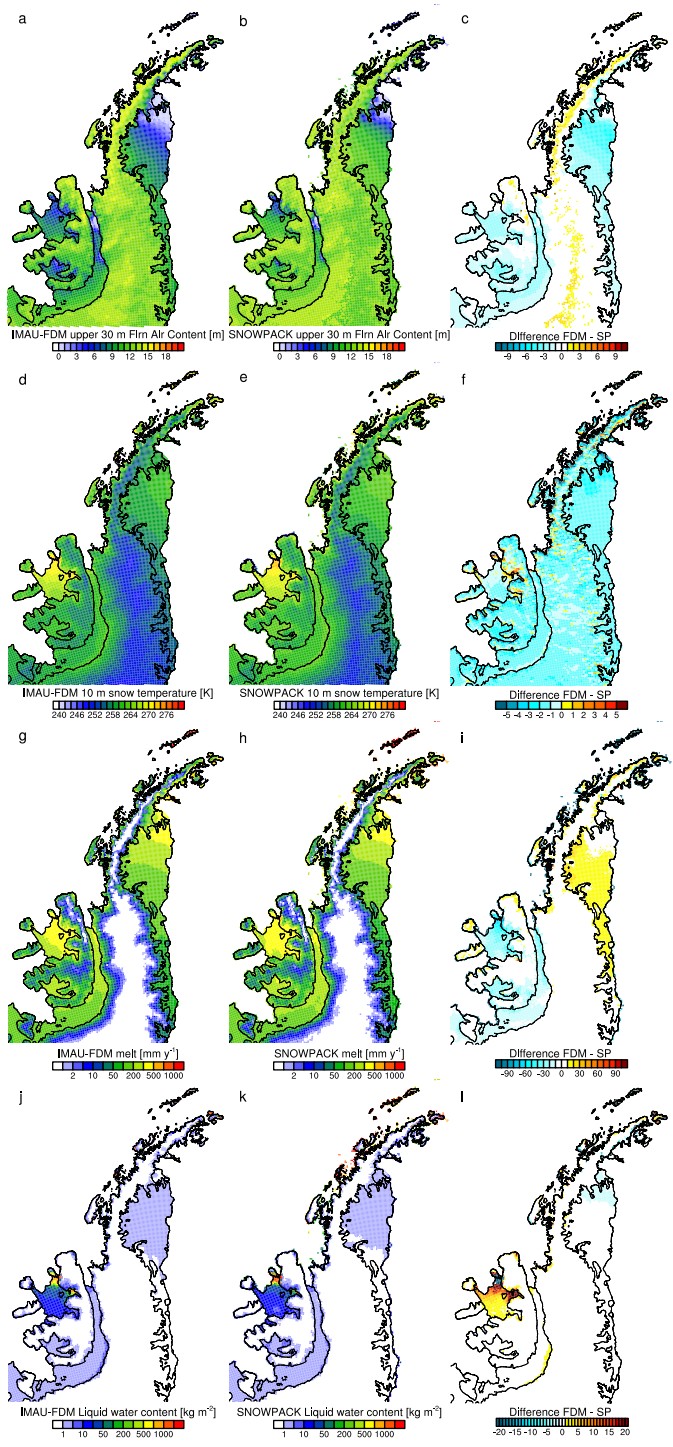

**Figure 4.** Modelled IMAU-FDM (left column), SNOWPACK (middle column) and their difference (IMAU-FDM - SNOWPACK) (right column) annual average (1979–2016) (a-c) Firn Air Content (FAC; m), (d-f) 10 m snow temperature (K), (g-i) surface meltwater production (mm w.e. $y^{-1}$) and (j-l) vertically integrated liquid water content (kg $m^{-2}$).

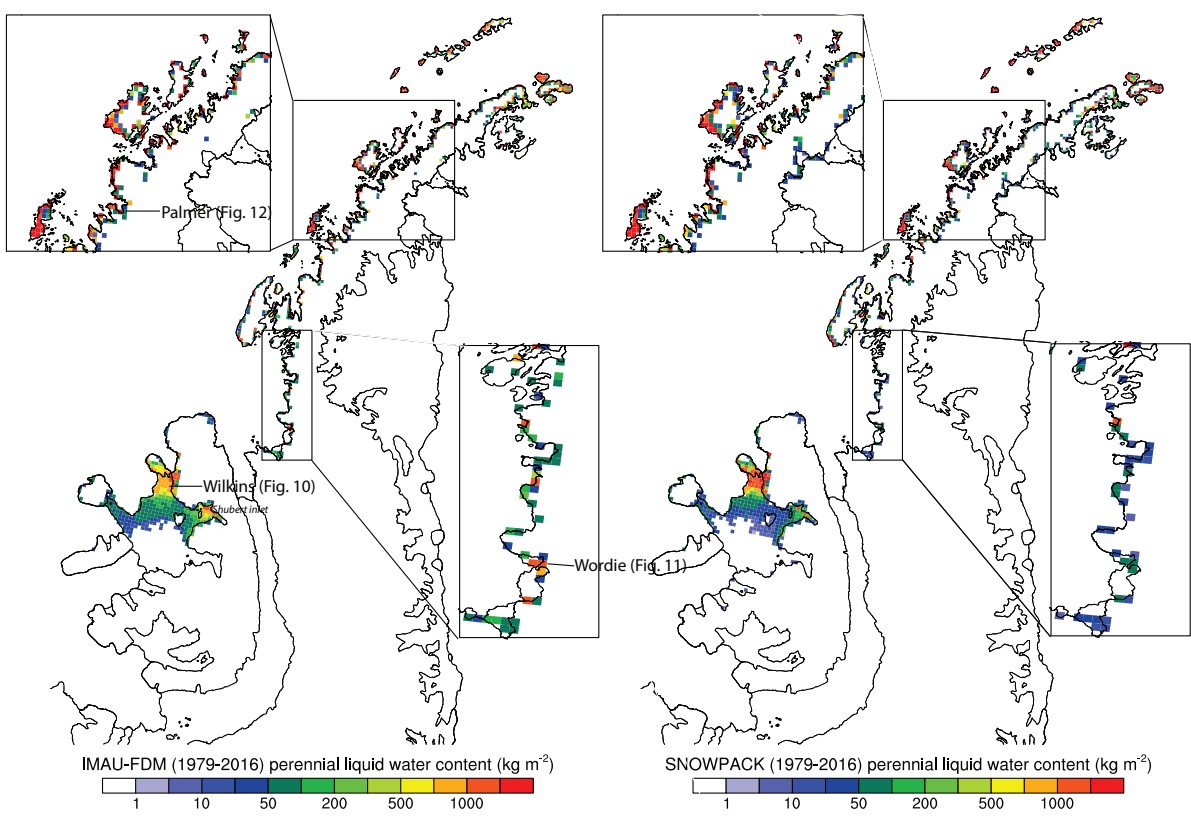

**Figure 5.** Modelled IMAU-FDM (left) and SNOWPACK (right) annual average (1979–2016) vertically integrated liquid water content for PFA points, see text.

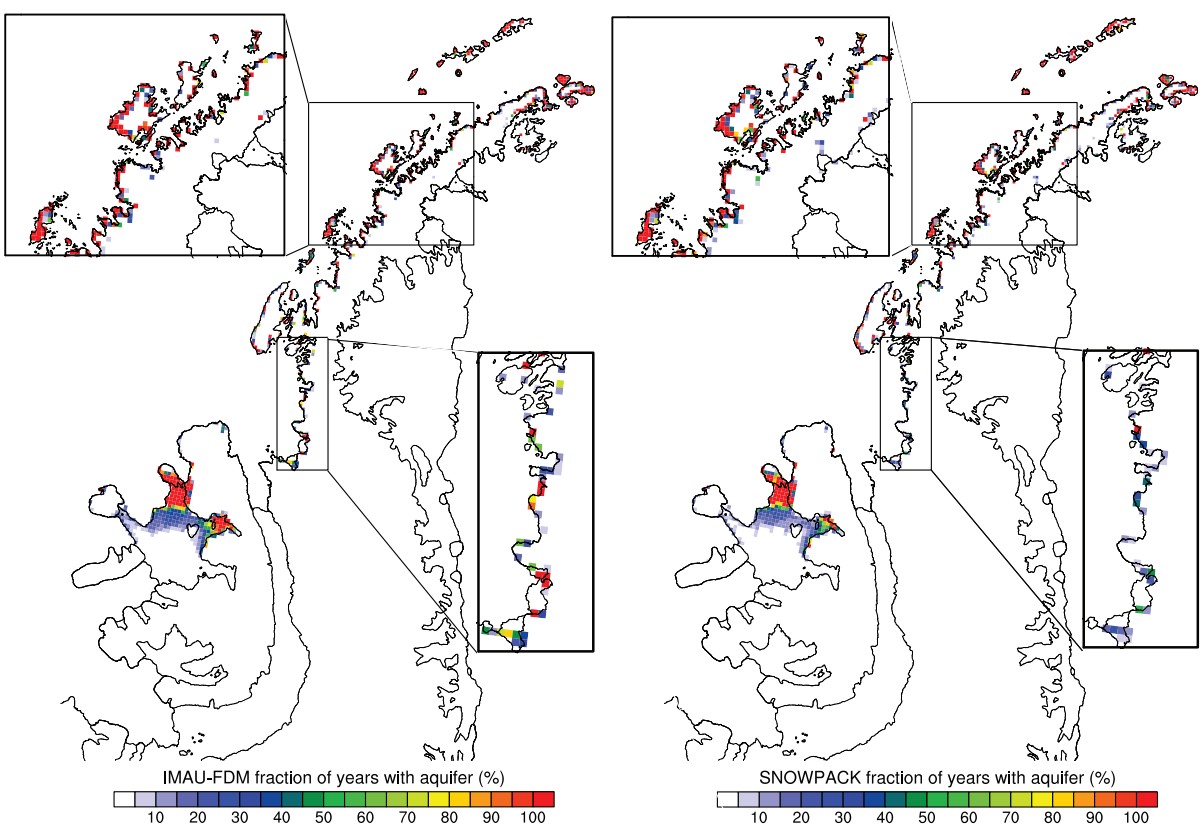

**Figure 6.** Modelled IMAU-FDM (left ) and SNOWPACK (right) percentage of total years from 1979–2016 with vertically integrated liquid water liquid water for PFA points, see text.

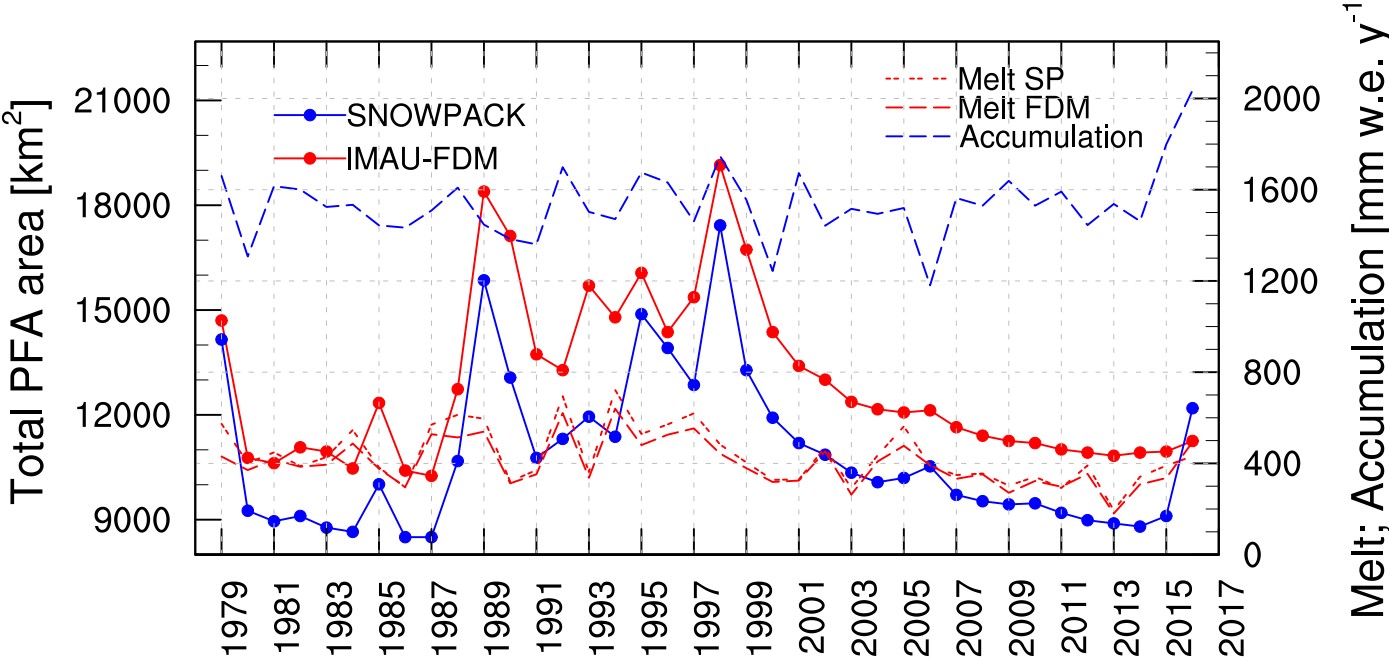

**Figure 7.** Yearly area (number of grid points multiplied by the gridbox area) of PFAs for SNOWPACK (blue solid line) and IMAU-FDM (red solid line) on the left axis, and yearly accumulation (blue dashed line) and melt (red dashed lines) averaged over all 941 PFA locations.

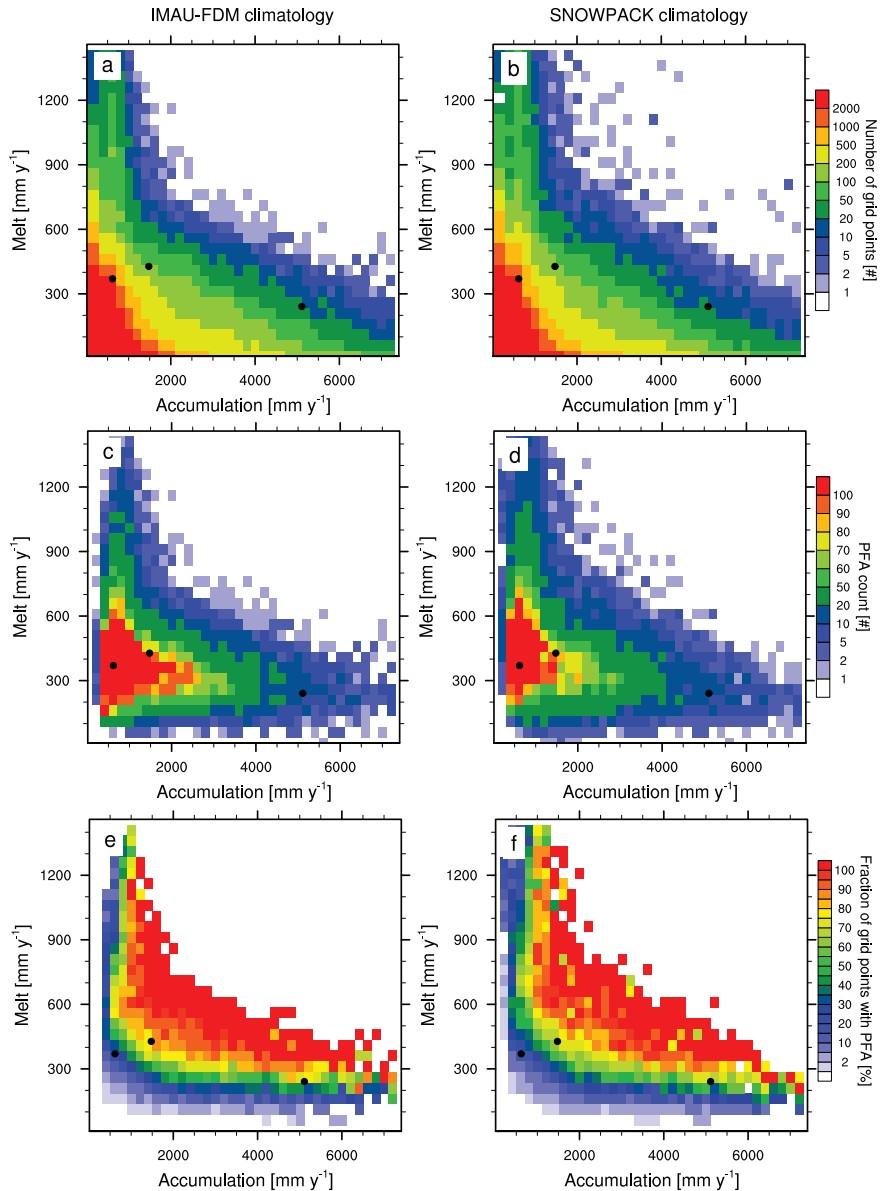

**Figure 8.** (a,b) Total number of model grid points in IMAU-FDM (left column) and SNOWPACK (right column), (c,d) total number of PFA occurrences for all model grid points in the respective hydrological year of the PFA, and (e,f) percentage of total number of PFA occurrences in (c,d) (w.r.t. Fig. 8a,b), as a function of yearly surface meltwater production (y-axis) and accumulation (x-axis). PFA numbers are binned in 50 mm w.e. $y^{-1}$ intervals in the y-direction and by 200 mm w.e. $y^{-1}$ in the x-direction. Grid points with only one occurrence are not shown.

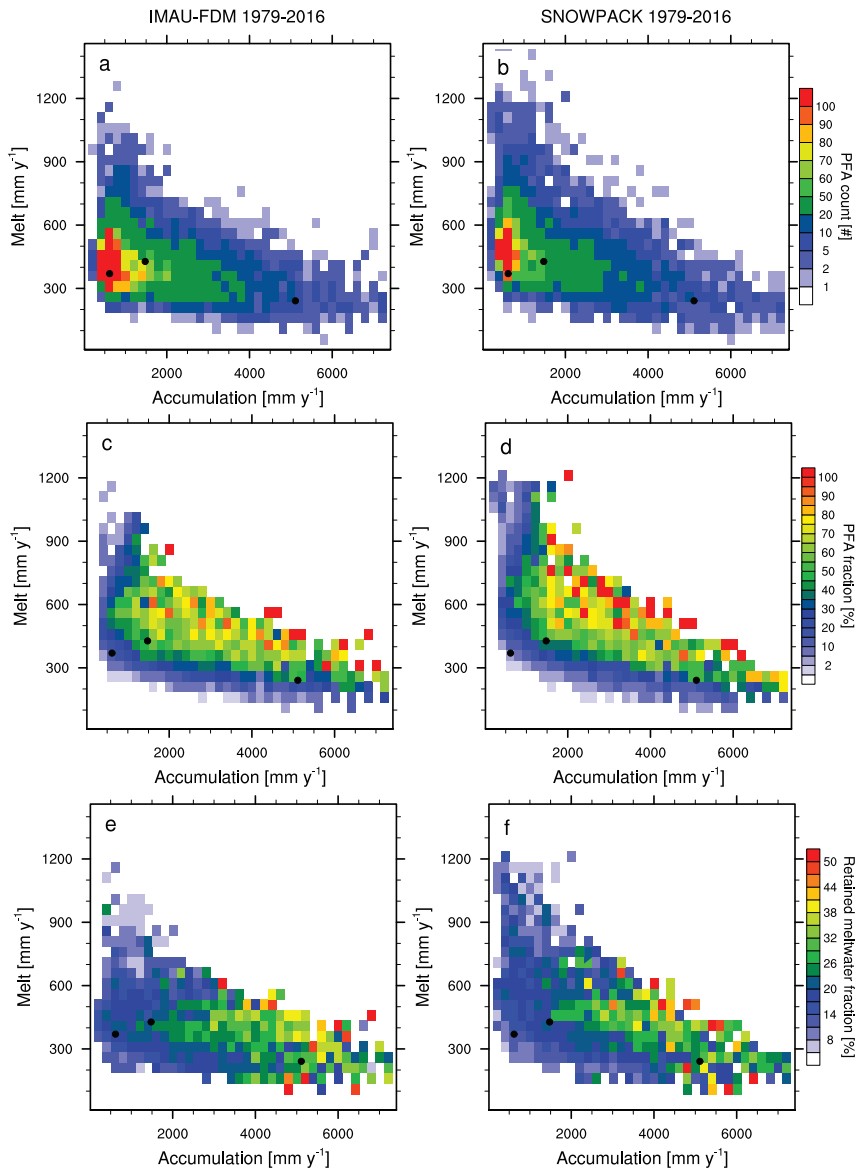

**Figure 9.** (a,b) Total number of PFA occurrences in IMAU-FDM (left column) and SNOWPACK (right column) of (increases in) perennial firn liquid water (as defined in text) for all model grid points in the respective hydrological year of the PFA, (c,d) percentage of total number of PFA occurrences (w.r.t. Fig. 8a,b), and (e,f) percentage of total produced meltwater that is retained in the firn at the end of the subsequent winter, as a function of yearly surface meltwater production (y-axis) and accumulation (x-axis). PFA numbers are binned in 50 mm w.e. $y^{-1}$ intervals in the y-direction and by 200 mm w.e. $y^{-1}$ in the x-direction. Grid points with only one occurrence are not shown.

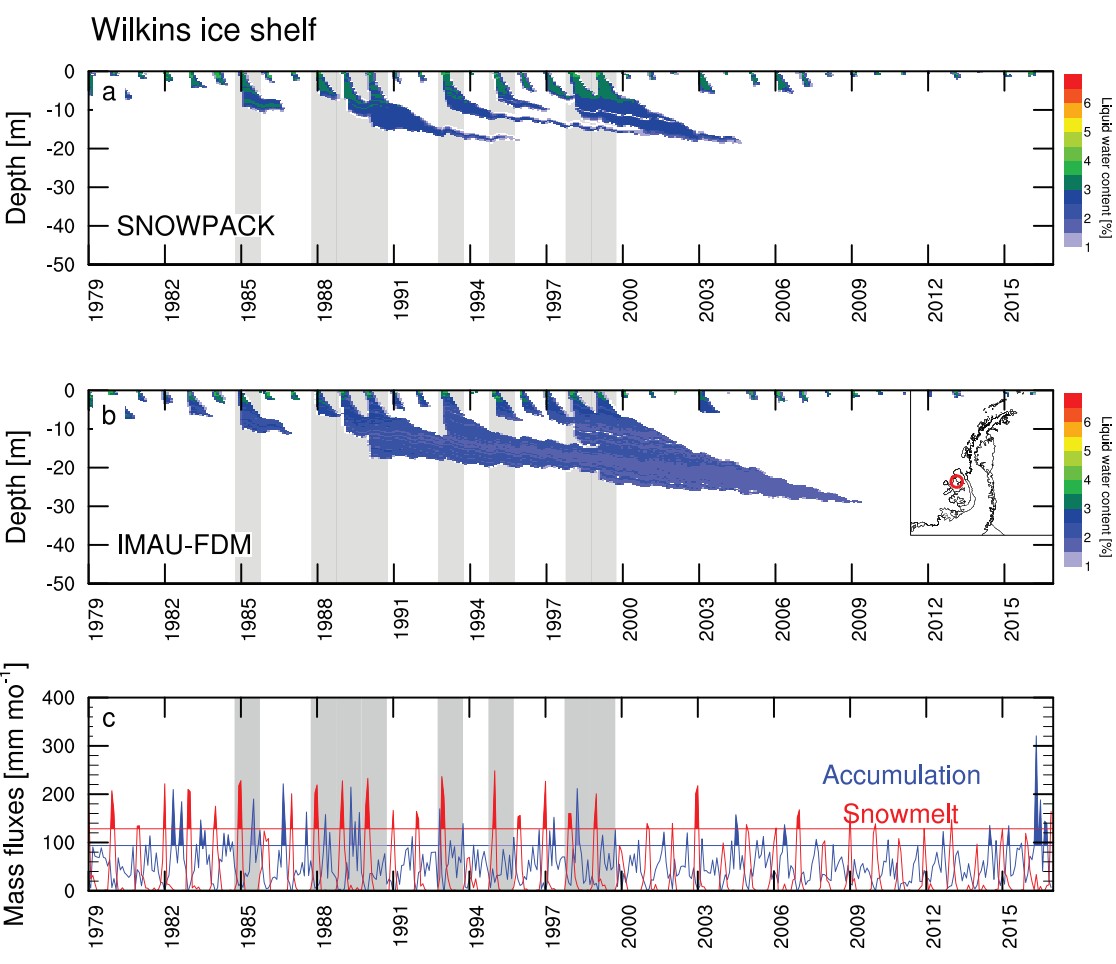

**Figure 10.** Volumetric water content for SNOWPACK (a) and IMAU-FDM (b) for 1979–2016 at a location on Wilkins ice shelf (see inset and Fig. 5). c) Monthly accumulation (blue) and surface meltwater production (red) for the same location; horizontal lines denote the mean + one standard deviation ($\sigma$) threshold. Grey bars denote hydrological years in which >5% of meltwater is retained. Only RACMO2 surface meltwater production is shown for clarity, which is similar to SNOWPACK values (see Fig. 7).

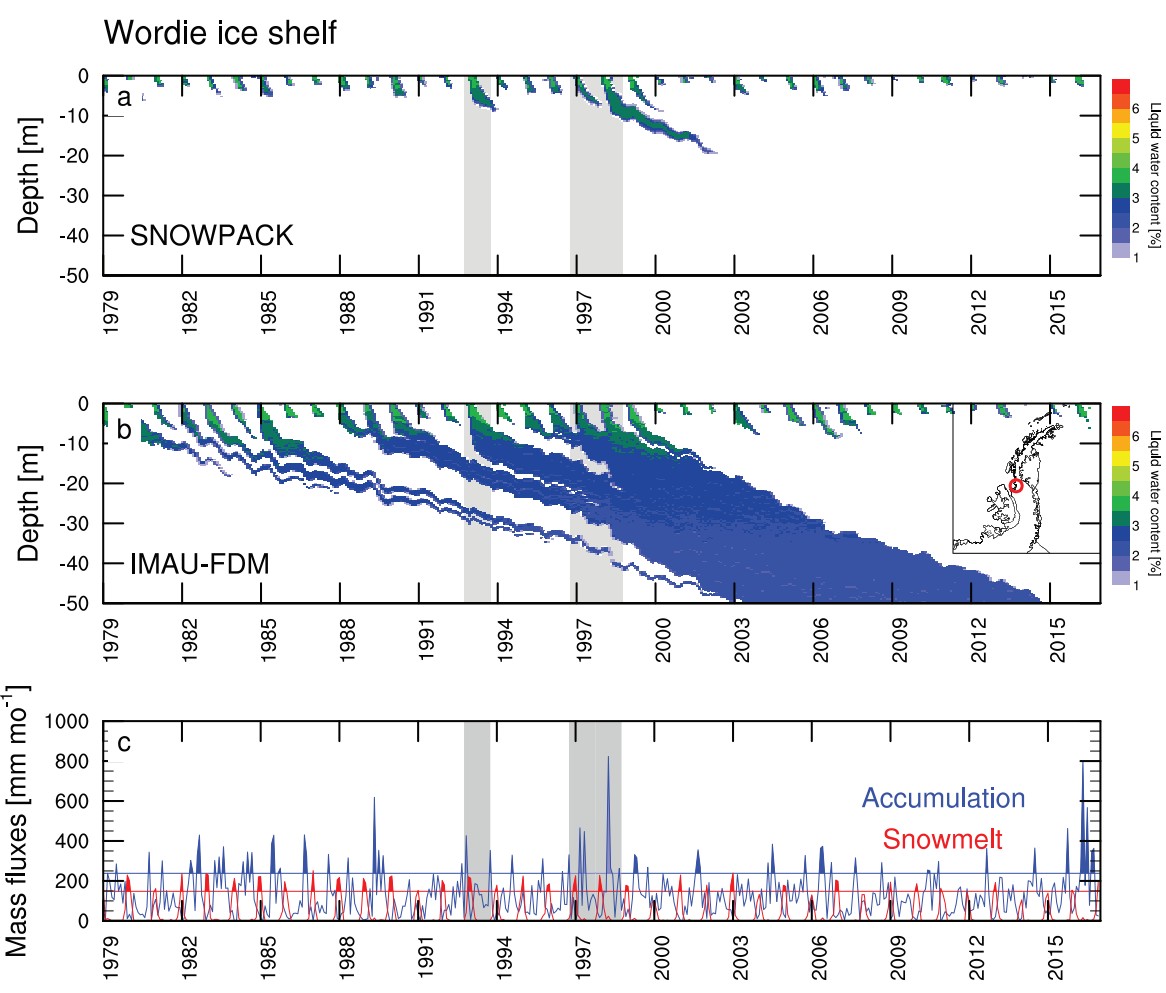

**Figure 11.** As for Fig. 9 but for a location on Wordie ice shelf (see inset and Fig. 5).

## Palmer Land

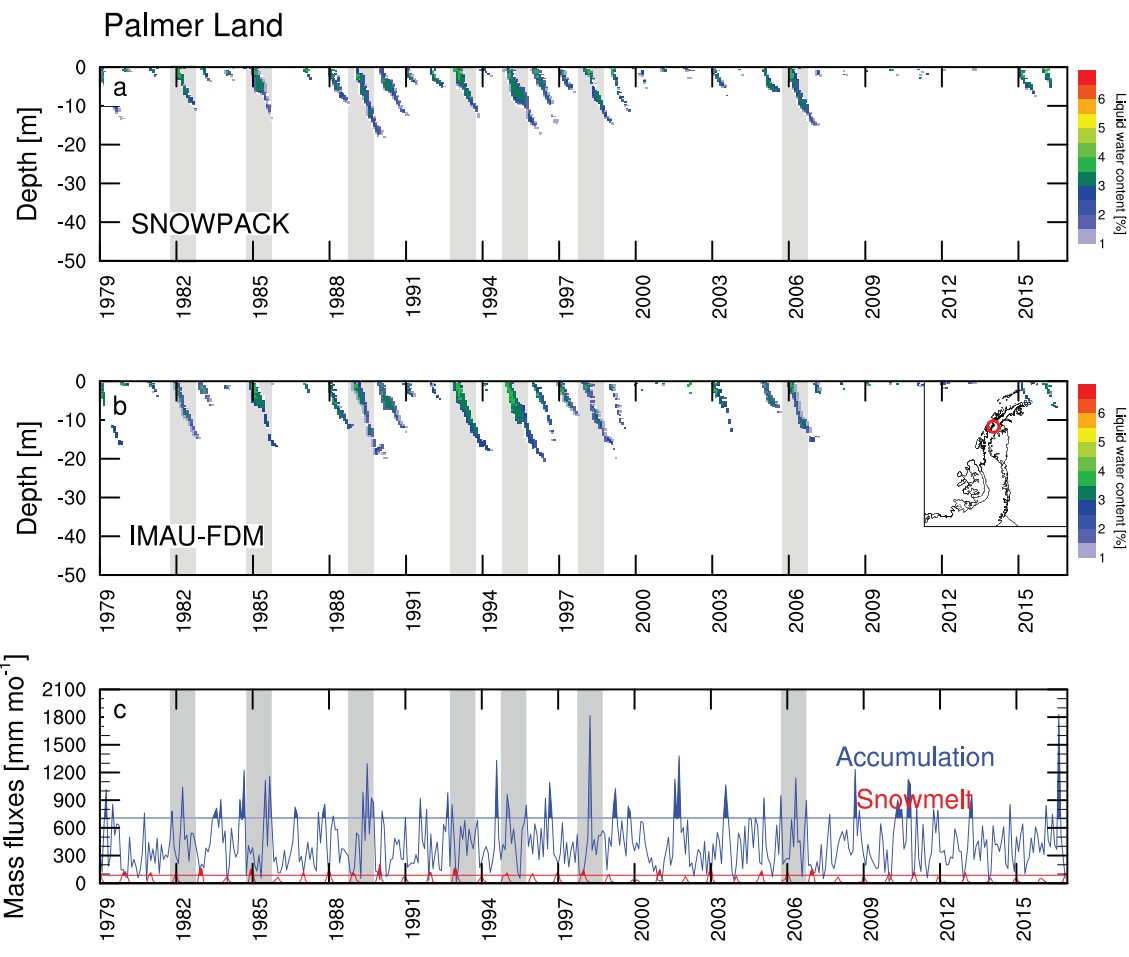

**Figure 12.** As for Fig. 9 but for a location in Palmer Land (see inset and Fig. 5).

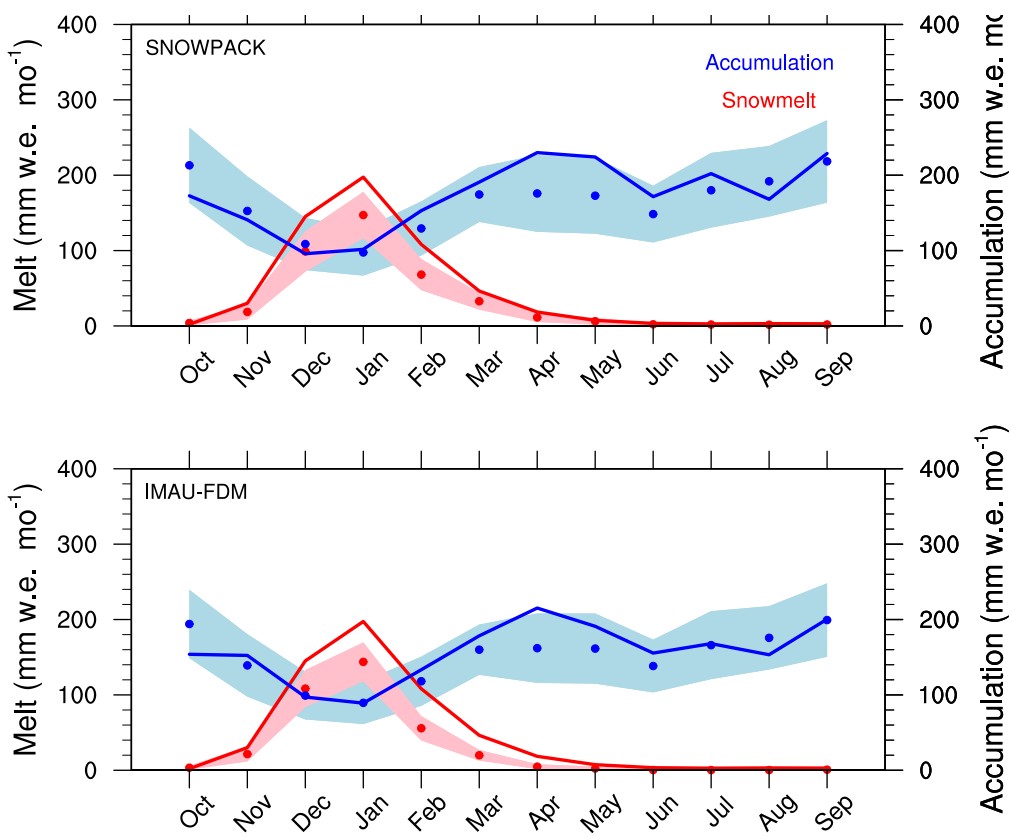

**Figure 13.** Monthly climatological (1979-2016) average surface melt production (left axis, red) and accumulation (right axis, blue) for years of >25% increases in perennial firn liquid water content. The solid lines show the monthly averages of formation years, while the dots show the monthly averages of all years in the 1979-2016 period.