# Peer review of "An exploratory modelling study of perennial firn aquifers in the Antarctic Peninsula for the period 1979-2016."

_The Cryosphere, 2020_

## Referee Comment (RC1) · Kristin Poinar (Referee) · 15 Aug 2020

Kristin Poinar (Referee)

kpoinar@buffalo.edu

**General comments**

This paper describes the first climate / SMB model study of perennial firn aquifer (PFA) presence in Antarctica, specifically the Antarctic peninsula. This analysis is timely and relevant. Currently, Antarctic melt is generally represented as refreezing immediately in the firn, running off directly into the ocean, or (in the most advanced representation) forming supraglacial lakes atop ice shelves. Recently, Lenaerts et al. (2016) and Dunmire et al. (2020) demonstrated that melt also persists within the firn, as buried lakes, in a specific region of Antarctica. This paper extends that discovery by demonstrating the great spatial extent of subsurface water (here, PFAs) along the Antarctic Peninsula.

[Figure]

The paper is well constructed, clearly written, and informative. The figures are strong and illustrate the points well. The analysis performed is fairly simple: spatiotemporally varying properties of the firn layer, including and culminating in its perennial liquid water content (PFA presence or absence), are compared between two snow models and to available data. Both models are forced by the same regional climate dataset over the same period. Both models have been fully developed in previous work and have also been applied, with success, to PFAs in Greenland (Steger et al., 2017a and 2017b). Both models are moderately complex, with detailed physical representation of depth-dependent snow properties, vertical water transport by the bucket scheme, and no horizontal water transport or ponding. The more advanced model, SNOWPACK, is run in a simpler mode for better comparison to IMAU-FDM. Each model has its strengths and weaknesses; the authors use two models to allow a more diverse approximation of reality, rather than to evaluate which is better.

This exploration teaches that PFA climates in Antarctica are broadly similar to those in Greenland, as found in previous model-based studies of Greenland PFAs by this group. This is the first such study of PFAs in Antarctica. It will be useful for scouting of potential field sites for campaigns on SMB or ice-shelf stability. Furthermore, this work sets up future study of potential links between PFAs and ice-shelf disintegration events, e.g. as hypothesized by the authors on the former Prince Gustav and Wordie Ice Shelves.

**Specific comments**

- P5 L14-15: "At colder locations, firn temperature is somewhat overestimated." The data on Figure 2 do not bear this out – the models underestimate $T_{10m}$ at all temperatures. Extrapolating the linear fits to colder temperatures makes the modeled temperature go above the 1:1 line, but this is not actually constrained by the data. I suggest removing this sentence.

- P9 L1-2: "it is likely that more PFAs will be formed" – this is not quite true. That conclusion would be based on Figures 9c-d, which showed no strong pattern for accumulation, as described in the previous paragraph. What should be said here is what Figures 9e-f show: that more meltwater will be retained in PFAs. This actually strengthens the conclusion here – higher melt influxes would increase rates of lateral water flow through PFAs, increasing the likelihood that hydrofractures drive to the bottom rather than refreeze / arrest partway (Poinar et al., 2017).

- Figure 11: The remarkable longevity of the IMAU-FDM aquifers on Wordie Ice Shelf (compared to the other ice shelves studied) suggests that they may have had a role in its collapse. Causation is far from certain, as is stated in the paper, but there is a potential mechanism: The persistence of the PFA for 10+ years (remarkable in this area) would increase the chance that lateral flow would bring the water to a crevasse, and the high LWC here (Figure 5) would increase the chance of deep hydrofracture, as in my above comment. This should be explicitly addressed in the manuscript, probably in Section 6.4, which is a little thin and would benefit from some mechanisms.

- P12-13 Section 7 and P1 Abstract: Many of the sentences in the abstract and conclusions section are very similar to one another. Both sections contain a lot of detailed facts related in somewhat choppy fashion. These sections should be smoothed and strengthened to better highlight the broad findings and important implications of this work, rather than specifics of its methods or more than a handful of quantitative results.

**Technical comments**

- Title: The parenthetical (1979-2016) in the title is odd and should be changed.

- P3 L22: I think the model time step is 3 hours (L27). For clarity, this should be stated here, rather than indirectly a few sentences later.

- P3 L29: Use "is" instead of "was"

- P4 L17: Use "is" instead of "was"

- Section 2.2 / 2.3: I suggest including the number of vertical layers in each model. I believe it is variable place to place depending on the firn properties; thus perhaps state the maximum and/or a typical number of layers in these runs.

- P5 L28: The abbreviations CI and WI are unclear and undefined here in the text. After a little hunting, I found them on Figure 1. I suggest referring to these cores by location and/or pointing back to Figure 1 when they are mentioned.

- P6 L3: "dry snow"

- P6 L21: "more quickly" instead of quicker

- P7 L32: Write out "1990s"

- P8 L1-8: These few sentences are confusing. Which model predicts a larger surface area of PFAs? I believe that the sentences contradict each other.

- P8 L26: I don't see this pattern or the threshold of <1500 mm we /yr

- P8 L27-28: Furthermore, the 100% points are sparse; their neighbors are more like 50-60%. This noise makes it even less likely that these 100% points are "real".

- P9 L8: The observations in Scambos et al. (2009) are satellite-based, not in situ

- P9 L18: "as temperatures decrease" – the surface temperature data are not shown, which is fine, but that should be briefly acknowledged. Alternately, they could be added to Figure 10c / 11c / 12c.

- P12 L 24-25: with with

- Figure 1: Add label for Wordie Ice Shelf. I eventually found it on Figure 5, but I expected it here.

- Figures 8,9: I very much like these figures.

**Review summary**

This is a good paper with structured organization, clear writing, and comprehensive figures. It connects climate to the subsurface hydrology of the Antarctic Peninsula, which is an important new step in understanding drivers of ice-shelf disintegration. It is a timely study that I enjoyed reading.

**References**

Dunmire, D. et al. (2020), Observations of buried lake drainage on the Antarctic Ice Sheet, Geophysical Research Letters, doi:10.1029/2020GL087970.

Poinar, K., et al. (2017), Drainage of Southeast Greenland Firn Aquifer Water through Crevasses to the Bed, Frontiers in Earth Science, doi:10.3389/feart.2017.00005.

---

## Referee Comment (RC2) · Anonymous Referee #2 · 18 Aug 2020

The manuscript by van Wessem et al. describes the formation and persistence of Antarctic Peninsula perennial firn aquifers in two models (IMAU-FDM and SNOW-PACK). Overall, both models show strong similarity in the location and timing of firn aquifer formation with minimal differences driven by various model parameters. These results suggest that careful consideration of model parameters and further investigation into firn processes are necessary to resolve the admittedly limited differences between the used models. However, the general similarity suggests that near surface hydrology has potentially played an important role in long-term ice sheet and ice shelf dynamics and further model development including meltwater transport is warranted.

Overall, the manuscript is generally reasonably written (see notes) and confirms many of the modeling and observation results initially developed in Greenland. Despite this,

there are several areas where the manuscript would benefit from additional improvements.

This manuscript is first and foremost a limited inter-model sensitivity study. While, I do believe it is relevant to more carefully and systematically examine the role of different model parameters in controlling firn aquifer formation and evolution. However, this is likely untenable at this point in the process. Therefore, it is strongly recommended that the manuscript very carefully expand the discussion on factors contributing to the differences in the models, with perhaps a table clearly laying out model differences and whether those differences are tunable/changeable, etc. Right now, it is only a few lines. But the authors can easily make an effort to better describe model differences (including irreducible water content) and how they would theoretically affect firn aquifer formation.

The manuscript makes a number of compromises that add to uncertainty in the results. These compromises may be warranted, but they do need to be clearly justified. First, the spin up of SNOWPACK leaves something to be desired. I understand issues with computational availability, but in light of this there needs to be a clear note as to how this with affect the inter-comparison with IMAU-FDM. Next, the comparison of the model results to firn cores from before the modeling window needs to be better scientifically justified – there are some notes to this point in the line comments. Finally, there is the issue of using RACMO melt for SNOWPACK, when it generates its own melt. Here, the argument presented is simplification of figures. This is a silly reason to make a change that will increase the uncertainty in the SNOWPACK results, and I strongly suggest that the authors reconsider this choice – it will make a minimal difference in the figures, but provide more accurate results.

The text could use a through tightening and focusing in some areas and expansion in others. This includes an improvement in the citations, which now are quite limited and narrowly focused on Antarctica; the clear separation of the results and discussion; and an overall focus on only the relevant components of the story. The results section has

quite a bit of discussion related material but lacks quantitively analysis of the results - there are some cases, like the range of conditions where firn aquifers form and persist where numbers would be beneficial to the community and future research. The discussion generally lacks a robust integration of these results within the current body of literature and instead seems to focus on things that need to be added to the models as suggested by others. In addition, the authors need to decide if the in-built RAMCO firn model is important to also consider and if so, the results need to be fully incorporated into the manuscript and figures.

Minor comments Page 1 4-5. What does adequately mean? Some quantitative assessment would be useful – even simply the direction of the bias in each model. 12. Quantify 'most' with a percentage. 12. Quantify 'large part' with a percentage. 16. The word 'timing' doesn't adequately describe what is meant. It's more like intra-annual variability or relative variability between SMB loss and gain. Consider more carefully framing this sentence more carefully and clearly. 24. The phrase 'as well as precipitation rates' should be changed to fit with the form of the other phrases in the sentence.

Page 2 9. There are tons more references to the discovery and behavior of GrIS firn aquifers. In the least, there should be a couple more citations and an 'e.g.' 11. The Bell paper is a perspective, while 'peer-reviewed', there are much better articles to cite here and, in the line below, including, but not limited to Bell et al. (2017 – Nature). 13. Latent heat release is only relevant when the FAs refreeze. This point should be clarified. I will also note again, that there is a broad body of literature about 'cryohydrologic warming' on the GrIS. 14. This line should simply be removed. It isn't necessary for the manuscript to be successful and frankly, unclear if it is true. There are AGU abstracts (which depending on the Journal, considered published, e.g. Miller et al., 2019 – AGU abstract 2019) and papers about supraglacial lakes and subglacial ponding also generally have discussion about firn aquifers. And, in all possibilities, there could easily be a paper in revision, review, press or published during the publication process of this manuscript. 20. This line somewhat implies that there is 'significant' melting during the

winter on the AP. Consider rephrasing. 26. 'observational datasets' across Antarctica or expand references. 26-27. So technically, Forster et al. (2014) utilize the firn model integrated with RACMO, which is, as indicated, different than the IMAU-FDM used in this study. And both citations use previous RACMO versions.

Page 3 4-5. The inclusion of the in-built firn model in RACMO isn't really justified or integrated into the discussion or abstract in any way. The authors should carefully consider whether it provides useful information. If so, it should be more clearly incorporated in the latter parts of the manuscript. 7. This section should focus on relevant atmospheric characteristics. The firn component should be included in the IMAU-FDM section, where it parses how the in-built model is different from the model primarily used in the manuscript. 23. 'Low' relative to what? Other models, observations?

Page 4. 18. Is the weaker densification due to chosen tunable variables, the used densification parametrization, or something else (or some combination)? 20. Perhaps mention that these are on the Plateau and do not affect the areas discussed herein (if this is the case). i.e. emphasize that the model didn't crash in areas analyzed in this paper. 26. Clarify this is because the forcing data is only available from 1979. 27. This statement (Using earlier. . .) should be expanded upon, essentially, this paragraph should clearly and strongly justify why using validation observations from a completely different timeframe can be used. Particularly in light of the recent, rapid atmospheric changes in the area. I think it is possible (e.g. at depth firn temperatures evolve slowly in response to surface forcings), but a careful, well-cited justification should be presented since this is the primary validation method of the manuscript.

Page 5. 10-11. r values should be accompanied by p-values or some measure of statistical significance. 12-13. The inclusion of the RACMO firn results are somewhat ad-hoc, either include them completely with a through discussion as to why the results are different from the other models or do not include them. 10-21. There are some discussion points here that should be moved to the Discussion. 23-4. There is a lot of better vs worse discussion here that heavily rely on Figure 3, which is simply a

qualitative comparison. Care should be taken to quantitively justify 'better' vs 'worse' statements.

Page 6. 1-4. These two sentences don't make a whole lot of sense and should be revised to be clearer and focus on the point of the manuscript. Plus, it is unclear what methodology the previous statement 'confirms' 5. This and the previous section title should be reconsidered. Something more descriptive like "Model characteristic inter-comparison" might be more useful in guiding the reader. 6-7. This should be clarified in the model descriptions above. 31. 'shelve' is the verb

Page 7. 9-10. It would be nice to have a volume comparison too, since there are observational estimates of GrIS PFA volume. 19-20. If they only last 1 year, are they really perennial? 20-22. See main note. 29. Missing an 'I' 30-31. I believe that this is an unnecessary simplification which introduces unnecessary and unaccounted for uncertainty. For the most part, this would mean 3 instead of 2 lines. See main note.

Page 8. 4-9. This is probably going to be a main take away from this manuscript. I would consider emphasizing this by placing the ratio of melt to accumulation on Figure 7. 5. This sentence indicates there is clear relation, but the previous sentence indicates that there is no clear relation. This conflict should be resolved. 7-10. This seems like a symptom of the different irreducible water content values in the models and should be discussed further here or elsewhere. 20. Top right corner isn't quite the right description. 31. What is positive?

Page 9. 2. Leave the speculation until the discussion. 8. Somewhat confused about this reference and Alley et al. (2018) in association with page 2, line 14. Generally, an aquifer is a water baring medium, so if the firn has liquid water stored, it is an aquifer.

Page 10. Inter-model differences: This section should be expanded to systematically assess the model differences as best as possible. In an ideal world, this would involve a sensitivity study, but because the focus of the manuscript is on model inter-comparison with single model runs, an effort should be made to clearly delineate the differences

and how the differences are related to model characteristics vs tuned parameters.

Page 11. 10. Reference for 'other regions as well'? 11. It reflects the lack of lateral water flow. The relevant mechanisms of heat transport and release should be discussed because this is not the only mechanism that could result in cold firn. 23. Odd reference location 30. It would be useful for readers to expand this section.

Page 12. 23-24. Citations may be relevant.

Figures Figure 1. Topography should also be mentioned in main text. Figure 2. See comment about in-built firn model inclusion. Figure 3. The figure text is too small to read. Either make the figure bigger or move the text to a table. The figure should also indicate visually what cores have high melt rates vs low melt rates. Figure 5. Would love to see a difference map too Figure 6. I like this figure, but it would also benefit from a difference map and a color bar that had both % and number of years (since both are used in the main text). Figure 7. I'd made this two (or 3) panels and also include the ratio of melt to ablation for both models over time Figure 8. Is the melt used here for SNOWPACK its actual melt or the RACMO melt? See previous point on this. The caption could also use some refinement to more clearly indicate what each of the panels is. Figure 11. What exactly is the shaded spread?

---

## Author Comment (AC1) · 27 Oct 2020

**Author replies (AC) to reviewer comments (RC) in blue, revised text in *"red italics"*.**

**REVIEWER #1**

**General comments**
This paper describes the first climate / SMB model study of perennial firn aquifer (PFA) presence in Antarctica, specifically the Antarctic peninsula. This analysis is timely and relevant. Currently, Antarctic melt is generally represented as refreezing immediately in the firn, running off directly into the ocean, or (in the most advanced representation) forming supraglacial lakes atop ice shelves. Recently, Lenaerts et al. (2016) and Dunmire et al. (2020) demonstrated that melt also persists within the firn, as buried lakes, in a specific region of Antarctica. This paper extends that discovery by demonstrating the great spatial extent of subsurface water (here, PFAs) along the Antarctic Peninsula. The paper is well constructed, clearly written, and informative. The figures are strong and illustrate the points well. The analysis performed is fairly simple: spatiotemporally varying properties of the firn layer, including and culminating in its perennial liquid water content (PFA presence or absence), are compared between two snow models and to available data. Both models are forced by the same regional climate dataset over the same period. Both models have been fully developed in previous work and have also been applied, with success, to PFAs in Greenland (Steger et al., 2017a and 2017b). Both models are moderately complex, with detailed physical representation of depth-dependent snow properties, vertical water transport by the bucket scheme, and no horizontal water transport or ponding. The more advanced model, SNOWPACK, is run in a simpler mode for better comparison to IMAU-FDM. Each model has its strengths and weaknesses; the authors use two models to allow a more diverse approximation of reality, rather than to evaluate which is better. This exploration teaches that PFA climates in Antarctica are broadly similar to those in Greenland, as found in previous model-based studies of Greenland PFAs by this group. This is the first such study of PFAs in Antarctica. It will be useful for scouting of potential field sites for campaigns on SMB or ice-shelf stability. Furthermore, this work sets up future study of potential links between PFAs and ice-shelf disintegration events, e.g. as hypothesized by the authors on the former Prince Gustav and Wordie Ice Shelves

AC: We want to thank Kristin Poinar for her positive and constructive comments, which have improved our manuscript. We address her comments one by one below.

**Specific comments**
RC: P5 L14-15: "At colder locations, firn temperature is somewhat overestimated." The data on Figure 2 do not bear this out – the models underestimate T10m at all temperatures. Extrapolating the linear fits to colder temperatures makes the modeled temperature go above the 1:1 line, but this is not actually constrained by the data. I suggest removing this sentence.

AC: Removed.

RC: P9 L1-2: "it is likely that more PFAs will be formed" – this is not quite true. That conclusion would be based on Figures 9c-d, which showed no strong pattern for accumulation, as described in the previous paragraph. What should be said here is what

Figures 9e-f show: that more meltwater will be retained in PFAs. This actually strengthens the conclusion here – higher melt influxes would increase rates of lateral water flow through PFAs, increasing the likelihood that hydrofractures drive to the bottom rather than refreeze / arrest partway (Poinar et al., 2017).

AC: This is a good point and we have rephrased this sentence: *"…it is likely that more meltwater will be retained in the firn."* The latter part about the hydrofracture is moved to the discussion, also on request of Reviewer #2.

RC: Figure 11: The remarkable longevity of the IMAU-FDM aquifers on Wordie Ice Shelf (compared to the other ice shelves studied) suggests that they may have had a role in its collapse. Causation is far from certain, as is stated in the paper, but there is a potential mechanism: The persistence of the PFA for 10+ years (remarkable in this area) would increase the chance that lateral flow would bring the water to a crevasse, and the high LWC here (Figure 5) would increase the chance of deep hydrofracture, as in my above comment. This should be explicitly addressed in the manuscript, probably in Section 6.4, which is a little thin and would benefit from some mechanisms.

AC: Another good point, also brought up by Reviewer #2, and we added this text to Discussion section 6.4 this section and added reference to Poinar et al., 2017.

RC: P12-13 Section 7 and P1 Abstract: Many of the sentences in the abstract and conclusions section are very similar to one another. Both sections contain a lot of detailed facts related in somewhat choppy fashion. These sections should be smoothed and strengthened to better highlight the broad findings and important implications of this work, rather than specifics of its methods or more than a handful of quantitative results.

AC: We did our best to enhance readability of the abstract and conclusion sections. We also added to the conclusion section a sentence related to hydrofracture potential, relating to the comment above.

**Technical comments**

RC: Title: The parenthetical (1979-2016) in the title is odd and should be changed.
AC: Changed.

RC: P3 L22: I think the model time step is 3 hours (L27). For clarity, this should be stated here, rather than indirectly a few sentences later.
AC: Changed.

RC: P3 L29: Use "is" instead of "was"
AC: Changed.

RC: P4 L17: Use "is" instead of "was"
AC: Changed.

RC: Section 2.2 / 2.3: I suggest including the number of vertical layers in each model. I believe it is variable place to place depending on the firn properties; thus perhaps state the maximum and/or a typical number of layers in these runs.
AC: We have now added to Section 2.2 that IMAU-FDM uses 3000 layers (which are not always active if the majority of the column consists of ice). SNOWPACK has a variable

number of layers, so we cannot state that number explicitly, information that has now also been added to Section 2.3.

RC: P5 L28: The abbreviations CI and WI are unclear and undefined here in the text. After a little hunting, I found them on Figure 1. I suggest referring to these cores by location and/or pointing back to Figure 1 when they are mentioned.
AC: We added a reference to Fig. 1 in the caption of Table 1.

RC: P6 L3: "dry snow"
AC: Changed.

RC: RC: P6 L21: "more quickly" instead of quicker
AC: Changed.

RC: P7 L32: Write out "1990s"
AC: Changed throughout.

RC: P8 L1-8: These few sentences are confusing. Which model predicts a larger surface area of PFAs? I believe that the sentences contradict each other.
AC: What we meant to say is that PFA area is more variable in time and space in SNOWPACK, i.e. from year to year more grid points have a PFA in SNOWPACK, but IMAU-FDM has the largest contiguous PFA surface area. We clarified as follows: "*It is also interesting that SNOWPACK simulates PFAs over a larger number of different grid points (864 vs 796) from year to year, while in every individual year IMAU-FDM has a larger total surface area (and total LWC) (see Fig. 7); apparently, SNOWPACK models some PFAs at variable locations that only last a single year, while IMAU-FDM does not, see e.g. the small PFAs in the Larsen A and Larsen B embayments.*"

RC: P8 L26: I don't see this pattern or the threshold of <1500 mm we /yr
AC: Agreed. Changed to: *"…apart from a small band of higher values around a melt rate of 550 mm per year"*.

RC: P8 L27-28: Furthermore, the 100% points are sparse; their neighbors are more like 50-60%. This noise makes it even less likely that these 100% points are "real".
AC: Rephrased to: *"Towards the top right, conditions are found with the highest fractions of PFA formation, ..."*

RC: P9 L8: The observations in Scambos et al. (2009) are satellite-based, not in situ
AC: Changed.

RC: P9 L18: "as temperatures decrease" – the surface temperature data are not shown, which is fine, but that should be briefly acknowledged. Alternately, they could be added to Figure 10c / 11c / 12c.
AC: We added a reference for clarification.

RC: P12 L 24-25: with with
AC: Corrected.

RC: Figure 1: Add label for Wordie Ice Shelf. I eventually found it on Figure 5, but I expected it here.
AC: Changed.

RC: Figures 8,9: I very much like these figures
AC: Thanks!

**Review summary**
RC: This is a good paper with structured organization, clear writing, and comprehensive figures. It connects climate to the subsurface hydrology of the Antarctic Peninsula, which is an important new step in understanding drivers of ice-shelf disintegration. It is a timely study that I enjoyed reading.
AC: Thanks!

**References**
Dunmire, D. et al. (2020), Observations of buried lake drainage on the Antarctic Ice Sheet, Geophysical Research Letters, doi:10.1029/2020GL087970.
Poinar, K., et al. (2017), Drainage of Southeast Greenland Firn Aquifer Water through Crevasses to the Bed, Frontiers in Earth Science, doi:10.3389/feart.2017.00005
We have added both studies to the reference list.

**REVIEWER #2**

**General comments**
The manuscript by van Wessem et al. describes the formation and persistence of Antarctic Peninsula perennial firn aquifers in two models (IMAU-FDM and SNOWPACK). Overall, both models show strong similarity in the location and timing of firn aquifer formation with minimal differences driven by various model parameters. These results suggest that careful consideration of model parameters and further investigation into firn processes are necessary to resolve the admittedly limited differences between the used models. However, the general similarity suggests that near surface hydrology has potentially played an important role in long-term ice sheet and ice shelf dynamics and further model development including meltwater transport is warranted. Overall, the manuscript is generally reasonably written (see notes) and confirms many of the modeling and observation results initially developed in Greenland. Despite this, there are several areas where the manuscript would benefit from additional improvements.
We want to thank the referee for his/her detailed comments, which have improved our manuscript.

**Major comments**
RC: This manuscript is first and foremost a limited inter-model sensitivity study. While, I do believe it is relevant to more carefully and systematically examine the role of different model parameters in controlling firn aquifer formation and evolution. However, this is likely untenable at this point in the process. Therefore, it is strongly recommended that the

manuscript very carefully expand the discussion on factors contributing to the differences in the models, with perhaps a table clearly laying out model differences and whether those differences are tunable/changeable, etc. Right now, it is only a few lines.

AC: Instead of a model study, we prefer calling this study exploratory, as we now stress explicitly in the title. A detailed description of model differences between IMAU-FDM and SNOWPACK has been provided in Steger et al. (2017), as now stated at the beginning of Sections 2 and 4, and we prefer to not repeat this in the form of an additional table. The difference in settings compared to Steger et al. (2017) mainly constitutes the surface mass balance forcing in SNOWPACK, which calculates surface temperature, sublimation and melt from the provided near-surface climate, rather than prescribing it directly from RACMO2. We now state this more explicitly and treat this more extensively in Section 2.3 (Data and Methods: SNOWPACK), Section 4 (Results: model intercomparison) and Section 6.1 (Discussion: intermodel differences).

RC: But the authors can easily make an effort to better describe model differences (including irreducible water content) and how they would theoretically affect firn aquifer formation.

AC: The impact of differences in prescribed irreducible water content is now addressed more extensively in Discussion section 6.1 on inter model differences: *"IMAU-FDM has a fixed irreducible water content of 2%, substantially lower than (the snow temperature dependent) irreducible water content in SNOWPACK, which averages ~4%. As a result, in IMAU-FDM water percolates to greater depths quicker, where it either refreezes or runs off (Steger et al., 2017a). Exceptions are locations where melt rates are sufficiently high to saturate the whole firn column, e.g. on the northern Wilkins ice shelf and the islands in the northwestern AP. On the one hand the lower irreducible water content of IMAU-FDM allows meltwater to spread out deeper into the firn, where it can more efficiently refreeze or runoff, such as in regions such as the Larsen A and B ice shelf embayments, while in SNOWPACK some meltwater still remains in the upper layers. On the other hand, in regions with moderate melt rates such as Wordie ice shelf, the larger irreducible water content in SNOWPACK likely causes more meltwater to be retained in the upper layers, where it can be more efficiently refrozen by the winter cold wave, resulting in the much smaller PFAs present. Therefore, the effects of differences in the representation of irreducible water are important but also subtle, depending on the interplay of local firn and atmospheric conditions."*

RC: The manuscript makes a number of compromises that add to uncertainty in the results. These compromises may be warranted, but they do need to be clearly justified. First, the spin up of SNOWPACK leaves something to be desired. I understand issues with computational availability, but in light of this there needs to be a clear note as to how this with affect the inter-comparison with IMAU-FDM.

AC: Indeed, SNOWPACK spinup does not always fully refresh the full firn layer, but as this is the case mainly in in dry and cold regions this has almost no effect on our results, as clarified in Section 2.3: *"The model is spun up in a fashion similar to the IMAU-FDM, by forcing the model with as many climatological periods (1979-2016) as needed to refresh the entire firn layer. In general, densification in SNOWPACK is slightly weaker than in IMAU-FDM (Steger et al., 2017a) and as a result SNOWPACK spinup for some cold and dry locations does not refresh the full firn layer. The effects on the results of this study are very small."* This is

further elaborated upon in the Discussion section, p. 11, l. 18-23: *"Additionally, the spinup of both models is different. The spin-up time of IMAU-FDM is made dependent on the yearly average snowfall and surface meltwater production, so as to rebuild the entire firn layer. For SNOWPACK the same approach is used, but, with firn densification being weaker in SNOWPACK, the spin up does not everywhere replace the entire firn column. Longer spinups were not performed due to computational costs. This results in some of the SNOWPACK initial firn profiles to be colder than in IMAU-FDM, having more potential for refreezing. However, this only affects a few locations, and mostly in dry and cold regions where PFAs do not form."*

RC: Next, the comparison of the model results to firn cores from before the modeling window needs to be better scientifically justified – there are some notes to this point in the line comments.
AC: Please see our answer to RC27.

RC: Finally, there is the issue of using RACMO melt for SNOWPACK, when it generates its own melt. Here, the argument presented is simplification of figures. This is a silly reason to make a change that will increase the uncertainty in the SNOWPACK results, and I strongly suggest that the authors reconsider this choice – it will make a minimal difference in the figures, but provide more accurate results.
AC: We agree with the reviewer that this added uncertainty is less than ideal, but it is a result of the settings used in all model runs and can therefore not be reversed. Fortunately, the differences in melt rates are small. We discuss this in Section 4: *"Figs. 4g-i show that melt rates are largely similar. IMAU-FDM generally predicts somewhat larger values (~25 mm w.e. yr-1) than SNOWPACK, except over George VI and Wilkins ice shelves, and near the former Prince Gustav ice shelf. These differences in melt are explained by differences in the turbulent heat fluxes (not shown), but overall the differences are small, i.e. less than 10%."*

RC: The text could use a through tightening and focusing in some areas and expansion in others. This includes an improvement in the citations, which now are quite limited and narrowly focused on Antarctica; the clear separation of the results and discussion; and an overall focus on only the relevant components of the story.
AC: We have tried to tighten and focus the text, and expanded the section on intermodel differences (Section 6.1). In the sections indicated by Reviewer #2 we have added and/or expanded the literature citations dealing with firn aquifers in Antarctica and other regions. The specific additions are provided below in the line-by-line comments.

RC: The results section has quite a bit of discussion related material but lacks quantitively analysis of the results -there are some cases, like the range of conditions where firn aquifers form and persist where numbers would be beneficial to the community and future research. The discussion generally lacks a robust integration of these results within the current body of literature and instead seems to focus on things that need to be added to the models as suggested by others. In addition, the authors need to decide if the in-built RACMO firn model is important to also consider and if so, the results need to be fully incorporated into the manuscript and figures.
AC: We tried to better separate results and discussion, and to make the results sections more quantitative. We have removed the section about seasonality, which made the MS

overly long. We retained the figure discussing seasonality, but as the last figure of the paper it now serves as an outlook figure, with suggestions for future work and research directions.

**Minor comments (per page)**

**Page 1**
RC: 4-5. What does adequately mean? Some quantitative assessment would be useful – even simply the direction of the bias in each model.
AC: To accommodate this comment, yet keep the abstract concise, we have expanded this sentence by: *"An evaluation using 75 snow temperature observations at 10 m depth and density profiles from 11 firn cores, shows that output of both snow models is sufficiently realistic to warrant further analysis of firn characteristics."*

RC: 12. Quantify 'most' with a percentage.
AC: We have changed this to "...*on 49% of the ice shelf area, in up to 100% (depending on the model) of the years in the 1979-2016 period."* We also added this percentage to the abstract, the conclusions, and to the results section.

RC: 12. Quantify 'large part' with a percentage.
AC: Please see above.

RC:16. The word 'timing' doesn't adequately describe what is meant. It's more like intra-annual variability or relative variability between SMB loss and gain. Consider more carefully framing this sentence more carefully and clearly.
AC: We are not sure how to interpret "SMB loss and gain", as SMB is positive everywhere, but to be more specific, we changed to *"…but also the timing of precipitation events relative to melt events".*

RC: 24. The phrase 'as well as precipitation rates' should be changed to fit with the form of the other phrases in the sentence.
AC: Changed.

**Page 2**
RC: 9. There are tons more references to the discovery and behavior of GrIS firn aquifers. In the least, there should be a couple more citations and an 'e.g.'
AC: We agree and we added reference to Miller et al. (2020), Miege et al. (2016), Miller (O) et al. (2014) and Brangers et al. (2020), and changed the sentence to: *"These so-called perennial firn aquifers (PFA) are extensive on the Greenland ice sheet where they were first discovered by Forster et al. (2013). They have been further studied during subsequent field campaigns, both using in situ (e.g. Koenig et al., 2014; Miller et al., 2017), seismic (e.g. Montgomery et al. 2017) and airborne/satellite measurements (e.g Miège et al., 2016; Brangers et al., 2020; Miller et al., 2020)."*

RC: 11. The Bell paper is a perspective, while 'peer-reviewed', there are much better articles to cite here and, in the line below, including, but not limited to Bell et al. (2017 – Nature).

AC: Agreed, and we now have (added) the following references: Fountain and Walder (1998), Bell and others (2018), Kingslake and others (2015).

RC: 13. Latent heat release is only relevant when the FAs refreeze. This point should be clarified. I will also note again, that there is a broad body of literature about 'cryohydrologic warming' on the GrIS.
AC: PFA's form and decay as a result of a balance between refreezing and meltwater supply, so latent heat release above/below the PFA is an important process. We have added some references and text to underline this: *"…through latent heat release if eventually refrozen (Fountain and Walder, 1998, Hubbard et al., 2016)"*.

RC: 14. This line should simply be removed. It isn't necessary for the manuscript to be successful and frankly, unclear if it is true. There are AGU abstracts (which depending on the Journal, considered published, e.g. Miller et al., 2019 – AGU abstract 2019) and papers about supraglacial lakes and subglacial ponding also generally have discussion about firn aquifers. And, in all possibilities, there could easily be a paper in revision, review, press or published during the publication process of this manuscript.
AC: We removed the sentence.

RC: 20. This line somewhat implies that there is 'significant' melting during the winter on the AP. Consider rephrasing.
AC: Actually, Kuipers Munneke et al. (2018) do show significant wintertime melt, up to 23% of the annual total for the years with observations. To better reflect this, we changed the sentence to *"…and regular pronounced, foehn-induced melt events occurring even in winter (Kuipers Munneke et al., 2018)"*.

RC: 26. 'observational datasets' across Antarctica or expand references.
AC: Changed.

RC: 26-27. So technically, Forster et al. (2014) utilize the firn model integrated with RACMO, which is, as indicated, different than the IMAU-FDM used in this study. And both citations use previous RACMO versions.
AC: We have now changed the sentence to: *"The models are forced by realistic atmospheric and surface conditions from the regional climate model RACMO2.3p2 for the period 1979–2016, which has been extensively evaluated with observational datasets, as have previous versions (Van Wessem et al., 2015, 2016,2018), and includes a snow model that is physically identical to IMAU-FDM (Ettema et al., 2010; Ligtenberg et al., 2011)."* Minor differences between the RACMO2 snow model and IMAU-FDM are provided in Section 2 of the manuscript.

**Page 3**
RC: 4-5. The inclusion of the in-built firn model in RACMO isn't really justified or integrated into the discussion or abstract in any way. The authors should carefully consider whether it provides useful information. If so, it should be more clearly incorporated in the latter parts of the manuscript.
AC: We have carefully checked this and can confirm that the results from the RACMO2 internal model and IMAU-FDM are highly similar and that the differences do not add

additional insights. In the revised MS, this is now mentioned specifically in the data-methods Section 2.

RC: 7. This section should focus on relevant atmospheric characteristics. The firn component should be included in the IMAU-FDM section, where it parses how the in-built model is different from the model primarily used in the manuscript.
AC: To address this, we kept the sentence in this section but reformulated to: *"RACMO2.3p2 includes a 100-layer firn model that calculates percolation, refreezing and runoff of liquid water (Ettema et al., 2010). The output of this internal firn model is not used in this study as the model is physically identical to IMAU-FDM, described below, but the latter runs at a higher vertical resolution (100 layers in RACMO2 versus 3000 layers in IMAU-FDM)."*

RC: 23. 'Low' relative to what? Other models, observations?
AC: The chosen value of 2% is low relative to values proposed in Coleou et al. (1998) and Lafaysse et al., (2017), which both used values of 4% of total pore volume. The new sentence now reads: *"The irreducible water content is set to a relatively low constant value of 2% of the pore volume, compared to the temperature dependent ~4% in SNOWPACK and in other studies (Coléou and Lesaffre, 1998; Lafaysse et al., 2017), allowing meltwater to efficiently percolate down to lower layers, mimicking processes such as piping and meltwater retention."*

**Page 4**
RC: 18. Is the weaker densification due to chosen tunable variables, the used densification parametrization, or something else (or some combination)?
AC: It involves differences in the coefficients used, and the fact that IMAU-FDM is a semi-empirical model while SNOWPACK uses overburden pressure, i.e. is more physically-based. SNOWPACK was developed to simulate seasonal snow where overburden pressures are modest. It does contain some tuning parameters, which have been tuned in Steger et al., 2017, to improve the agreement between modelled and observed densities over an ice sheet where overburden pressure is large. The weaker densification might be related to the tunable parameters in SNOWPACK, but further specific tuning would go beyond the scope of this study, and refer to (Steger et al., 2017).

RC: 20. Perhaps mention that these are on the Plateau and do not affect the areas discussed herein (if this is the case). i.e. emphasize that the model didn't crash in areas analyzed in this paper.
AC: Corrected.

RC: 26. Clarify this is because the forcing data is only available from 1979.
AC: Changed.

RC: 27. This statement (Using earlier...) should be expanded upon, essentially, this paragraph should clearly and strongly justify why using validation observations from a completely different timeframe can be used. Particularly in light of the recent, rapid atmospheric changes in the area. I think it is possible (e.g. at depth firn temperatures evolve slowly in response to surface forcings), but a careful, well-cited justification should be presented since this is the primary validation method of the manuscript.

AC: This is a fair point. To investigate this assumption further, and knowing that reanalysis products are unreliable before 1979, we resorted to observational datasets from manned meteorological observatories that extend further back in time, i.e. Faraday/Vernadsky (western AP) and Marambio (eastern AP). we formulated our findings as follows: *"To investigate the potential impact of the non-overlapping periods of observations and models, we resorted to data from two stations from the SCAR-READER dataset (Turner et al., 2004) that have temperature observations from before 1979, one representing the western AP (Faraday/Vernadsky) and one the eastern AP (Marambio). Annual modelled temperature for 1979-2016 agree with RACMO2 within 0.35 K when compared to the overlapping period. The station temperatures are 1.2 and 0.5 K lower for the 1950-1978 period compared to the 1979-2016 period, a change representative of the warming in the 2nd half of the 20th century. We thus conclude that the generally underestimated model temperatures (see Fig. 2, next section) cannot be (partly) ascribed to the non-overlapping period."*

**Page 5**
RC: 10-11. r values should be accompanied by p-values or some measure of statistical significance.
AC: We now include a Table for the density profile statistics. All r values are with p<0.0001 as stated in the Table caption.

RC: 12-13. The inclusion of the RACMO firn results are somewhat ad-hoc, either include them completely with a through discussion as to why the results are different from the other models or do not include them.
AC: We removed all RACMO2 subsurface model results from the manuscript.

RC: 10-21. There are some discussion points here that should be moved to the Discussion.
AC: We prefer to use the Discussion section for PFA occurrence and its implications, rather than model quality and differences. That is why we decided to retain the brief interpretation of the temperature bias here.

RC: 23-24. There is a lot of better vs worse discussion here that heavily rely on Figure 3, which is simply a qualitative comparison. Care should be taken to quantitively justify 'better' vs 'worse' statements.
AC: We shortened this section by removing some of the qualitative comparisons, and retaining only the important differences. We also added a Table listing all statistics, and merged the evaluation into a single section.

**Page 6**
RC: 1-4. These two sentences don't make a whole lot of sense and should be revised to be clearer and focus on the point of the manuscript. Plus, it is unclear what methodology the previous statement 'confirms'
AC: These sentences have been reformulated and moved to the Discussion section.

RC: 5. This and the previous section title should be reconsidered. Something more descriptive like "Model characteristic inter-comparison" might be more useful in guiding the reader.
AC: Changed to "*Results: model intercomparison".*

RC: 6-7. This should be clarified in the model descriptions above.

AC: We reformulated this sentence: *"As described in the Section 2, the surface forcing of both models is different"*. In Section 2.3 we elaborate on this.

RC: 31. 'shelve' is the verb

AC: Corrected.

**Page 7**

RC: 9-10. It would be nice to have a volume comparison too, since there are observational estimates of GrIS PFA volume.

AC: In the current model settings, water storage in the firn is determined solely by the irreducible water content, i.e. the settings do not allow for water ponding on top of impermeable ice. A direct comparison with observed water volume, if available, would therefore not be viable. Still, because irreducible water that does not refreeze is a requirement for PFA formation, aquifer location, formation and horizontal extent can still be predicted with the current model settings, as has been demonstrated for the Greenland situation. We have rephrased a sentence in the Discussion section 6.2 to clarify this: *"As a result, a direct comparison with observed water volume is currently not viable, and the current approach should be regarded as an exploratory study. Nonetheless, because irreducible water that does not refreeze is a requirement for PFA formation, aquifer location, formation and horizontal extent can still be predicted using these simplified models (Ligtenberg et al., 2011; Forster et al., 2014; Steger et al., 2017a)."*

RC: 19-20. If they only last 1 year, are they really perennial?

AC: No aquifer will last 'forever', and that is why we decided to use as a definition of "perennial" ("long-lasting") as "lasting at least through the winter", i.e. up to the next melt season. Hence, to qualify as PFA in this study, liquid water must be present for at least one year.

RC: 20-22. See main note.

AC: Please see previous answer.

RC: 29. Missing an 'I'

AC: Corrected.

RC: 30-31. I believe that this is an unnecessary simplification which introduces unnecessary and unaccounted for uncertainty. For the most part, this would mean 3 instead of 2 lines. See main note.

AC: To address this comment, we updated Figures 8, 9 and 13 with the correct melt forcing. We left Figs. 10,11 and 12 unchanged as this would negatively impact their clarity, while leaving the conclusions unchanged. The figure captions have been adjusted.

**Page 8**

RC: 4-9. This is probably going to be a main take away from this manuscript. I would consider emphasizing this by placing the ratio of melt to accumulation on Figure 7.

AC: We have tried to create this figure, but it does not give a clear relation in any way we try to visualize it. We therefore decided to keep the Figure as is, but have rephrased the paragraph as follows: *"In Fig. 7, a clear relation between PFA extent and annual averages of accumulation or snowmelt is not obvious, but periods of PFA area growth do occur when average melt exceeds 400 mm w.e. y-1, or about 25% of the annual average accumulation, such as in 1986-1989 or 2005-2006."*

RC: 5. This sentence indicates there is clear relation, but the previous sentence indicates that there is no clear relation. This conflict should be resolved.
AC: Corrected.

RC: 7-10. This seems like a symptom of the different irreducible water content values in the models and should be discussed further here or elsewhere.
AC: We have reformulated the sentence as follows, not yet mentioning irreducible water content (see below): *"It is also interesting that SNOWPACK simulates PFAs over a larger number of different grid points (864 vs 796) from year to year, while in every individual year IMAU-FDM has a larger total surface area (and total LWC) (see Fig. 7); apparently, SNOWPACK models some PFAs at variable locations that only last a single year, while IMAU-FDM does not, see e.g. the small PFAs in the Larsen A and Larsen B embayments."*
The impact of differences in prescribed irreducible water content is now addressed more extensively in Discussion section 6.1 on inter model differences: *"IMAU-FDM has a fixed irreducible water content of 2%, substantially lower than (the snow temperature dependent) irreducible water content in SNOWPACK, which averages ~4%. As a result, in IMAU-FDM water percolates to greater depths quicker, where it either refreezes or runs off (Steger et al., 2017a). Exceptions are locations where melt rates are sufficiently high to saturate the whole firn column, e.g. on the northern Wilkins ice shelf and the islands in the northwestern AP. On the one hand the lower irreducible water content of IMAU-FDM allows meltwater to spread out deeper into the firn, where it can more efficiently refreeze or runoff, such as in regions such as the Larsen A and B ice shelf embayments, while in SNOWPACK some meltwater still remains in the upper layers. On the other hand, in regions with moderate melt rates such as Wordie ice shelf, the larger irreducible water content in SNOWPACK likely causes more meltwater to be retained in the upper layers, where it can be more efficiently refrozen by the winter cold wave, resulting in the much smaller PFAs present. Therefore, the effects of differences in the representation of irreducible water are important but also subtle, depending on the interplay of local firn and atmospheric conditions."*

RC: 20. Top right corner isn't quite the right description.
AC: Changed to *"...towards the top right corner..."*.

RC:31. What is positive?
AC: We changed this to *"Only when this fraction is positive…"*.

Page 9.
RC: 2. Leave the speculation until the discussion.
AC: Changed

RC: 8. Somewhat confused about this reference and Alley et al. (2018) in association with page 2, line 14. Generally, an aquifer is a water baring medium, so if the firn has liquid water stored, it is an aquifer.

AC: We removed the inference about aquifers.

**Page 10**

RC: Page 10. Inter-model differences: This section should be expanded to systematically assess the model differences as best as possible. In an ideal world, this would involve a sensitivity study, but because the focus of the manuscript is on model inter-comparison with single model runs, an effort should be made to clearly delineate the differences and how the differences are related to model characteristics vs tuned parameters.

AC: We expanded this section, e.g. with a discussion on differences in irreducible water content. See also our previous answers to major comments.

**Page 11**

RC: 10. Reference for 'other regions as well'?

AC: We changed this to: *"…which may apply to other regions with large melt rates as well, such as the embayments of Larsen A and B ice shelves."*

RC: 11. It reflects the lack of lateral water flow. The relevant mechanisms of heat transport and release should be discussed because this is not the only mechanism that could result in cold firn.

AC: Changed.

RC: 23. Odd reference/location

AC: To clarify the use of this citation, we added: *"…because this determines to a large extent how much meltwater is produced and the potential to fill pore space in the firn"*.

RC: 30. It would be useful for readers to expand this section.

AC: We expanded this section by adding the following, also based in the comment by Reviewer #1: *"Moreover, the remarkable longevity of the Wordie ice shelf PFA (at least as suggested by IMAU-FDM, Figs. 6 and 11) would increase the probability that lateral flow brought meltwater to a crevassed section of the ice shelf, increasing the likelihood of hydrofracturing (Poinar et al., 2017). Other disintegrated ice shelves in the eastern AP also show the potential for PFA presence on their grounding lines, e.g. former Prince Gustav and Larsen A ice shelves. This does not imply causation and could simply be a result of warmer conditions. In future work, the potential role of PFAs on ice shelf stability will be studied in more detail, to gain better understanding the fate of ice shelves in a warming climate in which both melt and snowfall, and thus PFA formation, are expected to increase."*

**Page 12**

RC: 23-24. Citations may be relevant.

AC: We have added references to Kingslake et al., 2015, Poinar et al. 2017 and Bell et al., 2018, the latter as a relevant literature review.

**Figures**

RC: Figure 1. Topography should also be mentioned in main text.

AC: Corrected.

RC: Figure 2. See comment about in-built firn model inclusion.

AC: See previous answers, we removed all RACMO2 firn model results from the manuscript.

RC: Figure 3. The figure text is too small to read. Either make the figure bigger or move the text to a table. The figure should also indicate visually what cores have high melt rates vs low melt rates.

AC: We moved the numerical information to a separate table. Melt rates are in the same range (150-300 mm w.e) for all cores, as they are located relatively close to each other. Hence, we think it is not necessary to reorganize the figure.

RC: Figure 5. Would love to see a difference map too. Figure 6. I like this figure, but it would also benefit from a difference map and a color bar that had both % and number of years (since both are used in the main text).

AC: We constructed such a difference plot (see below) but decided to not include it in the manuscript. Ideally this extra Figure should be a third panel in Figures 5 and 6, but that degrades the clarity of these figures, in which we maximized the visibility by only using two panels. As we feel that no significant new insights are gained from the below figure we decided to not include it as a separate figure either, also because the MS is already rich in figures as is. We also decided to keep the percentages, as it easier to interpret than number of years. We clarified in the text that 100% equals 37 out of 37 years.

RC: Figure 7. I'd made this two (or 3) panels and also include the ratio of melt to ablation for both models over time Figure 8. Is the melt used here for SNOWPACK its actual melt or the RACMO melt? See previous point on this. The caption could also use some refinement to more clearly indicate what each of the panels is.

AC: We decided not to expand this figure for reasons similar to those outlined above.

RC: Figure 11. What exactly is the shaded spread?

AC: Do you mean the grey bars? These highlight the PFA years where more than 5% of meltwater is retained (in both models), see the caption of Fig. 13.

[Figure]

Perennial LWC difference (IMAU-FDM - SNOWPACK)  (kg m$^{-2}$)

-100  -80  -60  -40  -20  0  20  40  60  80  100

Palmer (Fig. 12)

Wilkins (Fig. 10)

Wordie (Fig. 11)

PFA years difference (IMAU-FDM - SNOWPACK) (%)

-50  -40  -30  -20  -10  0  10  20  30  40  50

---

## Author Response (AR2)

Author replies (AC) to editor comments in blue, revised text in "red italics".

EC: I value the work done on the revised manuscript, which has appropriately addressed the reviewers' comments. However, the simulated aquifer spatial extents over the Wilkins ice shelf seem largely underestimated when compared to observations recently published by Montgomery et al. (2020). Montgomery et al. (2020) have mapped the aquifer extent on the Wilkins ice shelf using airborne radar observations, and they have confirmed the aquifer presence with ground-based radar data and drillings. While this study is referenced in your manuscript, there is no mention of this underestimation. As a result, I am returning this manuscript to you for adding clarification text that would benefit all readers.

AC: Dear editor. Thank you for your positive comments about our revision. We are aware of this new study having been released during (and after) the review and revision of this manuscript. We have tried to address your suggestions appropriately by adding the following paragraph to section 5.1.1.
"The presence of an extensive PFA on Wilkins ice shelf (WIS) as reported by Montgomery et al. (2020) is confirmed by our results. Although they do not provide a quantitative estimate of total surface area of the WIS aquifer, their results from the MCoRDS radar system onboard NASA's Operation IceBridge fight on 16 November 2014 suggest a greater westward aquifer extent than in either IMAU-FDM or SNOWPACK (Fig. 5). In addition, their study provides field observations of e.g. liquid water content and firn density. A comparison of these observations with our results remains difficult, as these in-situ observations were performed in 2018, i.e. after the model period used here. Moreover, neither firn model treats meltwater ponding on top of ice lenses or lateral water flow, further hampering such a direct comparison. It is well possible that the underestimated PFA extent is a result of these model limitations."

More specifically:
EC: 1) it appears that the simulated aquifer spatial extents (based on two snow models) presented in the current TCD manuscript do not match the large extent identified with observations based on an established airborne radar system. This is currently not stated in the manuscript in spite of a dedicated subsection to Wilkins (section 5.1.1). Also, the abstract may offer up front a misleading statement with the following sentence: "Most persistent and extensive are PFAs modelled on and around Wilkins ice shelf. Here, both meltwater production and accumulation rates are sufficiently high to cause PFA formation on 49% of the ice shelf area [...]". Based on airborne observations (and acknowledging that data only exists along track), it seems that a much larger area of the Wilkins ice shelf has an aquifer. The latter sentence should restate that the information provided results from simulations. Moreover, it seems appropriate to address this discrepancy thoroughly in section 5.1.1, because the simulated extent seems largely underestimated.
AC: As stated above, we have added a paragraph addressing this study in more detail in section 5.1.1. To accommodate the above comment, we have edited the sentence in the abstract: "Here, both meltwater production and accumulation rates are sufficiently high to sustain a PFA on 49% of the ice shelf area, in (up to) 100% (depending on the model) of the years in the 1979--2016 period. Although this PFA presence is confirmed by recent observations, its extent in the models appears underestimated." and in the conclusion section: "The most extensive PFA system is modelled on Wilkins ice shelf, covering 49% of its total area in both models. Observations confirm the presence of an extensive aquifer on the WIS, but suggest that the modelled extent may be underestimated (Montgomery et al., 2020)."

EC: 2) of note, Montgomery et al. (2020) have published density profiles on the Wilkins ice shelf, where there is an aquifer. The in situ, ground-based, and airborne measurements presented in that study are available at https://www.usap-dc.org/view/dataset/601390. I encourage the authors to add these new density profiles in their density model evaluation, which at the moment exclusively includes density profiles over non-aquifer areas on the Larsen C ice shelf.

AC: A direct comparison with these observations of liquid water content and density would require a hydrological model that explicitly treats meltwater ponding on top of ice lenses as well as lateral water flow. As neither firn model includes parametrizations of these processes, a direct comparison is deemed less useful, as we have now explained in the added text (see above).

**Reference**
Montgomery, L., Miège, C., Miller, J., Scambos, T. A., Wallin, B., Miller, O., et al. (2020). Hydrologic properties of a highly permeable firn aquifer in the Wilkins Ice Shelf, Antarctica. Geophysical Research Letters, 47, e2020GL089552. https://doi.org/10.1029/2020GL089552